# Adaptive Code Watermarking Through Reinforcement Learning

**Zhimeng Guo** [1]  **Huaisheng Zhu** [1]  **Siyuan Xu** [1]  **Hangfan Zhang** [1]  **Teng Xiao** [2,3]  **Minhao Cheng** [1]

## Abstract

As LLMs increasingly generate production code, protecting intellectual property demands watermarking techniques that respect code's strict syntactic constraints. In this work, we introduce CodeTracer, an innovative adaptive code watermarking framework underpinned by a reinforcement learning training paradigm. At its core, CodeTracer features a policy-driven approach that utilizes a parameterized model to intelligently bias token choices during next-token prediction. This strategy ensures that embedded watermarks maintain code functionality while exhibiting subtle yet statistically detectable deviations from typical token distributions. To facilitate policy learning, we devise a comprehensive reward system that seamlessly integrates execution feedback with watermark embedding signals, balancing process-level and outcome-level rewards. To enable gradient-based optimization of these discrete watermarking decisions, we employ Gumbel Top-k reparameterization. Extensive comparative evaluations demonstrate that CodeTracer outperforms state-of-the-art baselines across multiple benchmarks in both watermark detectability and code functionality. Our code is available at `https://github.com/TimeLovercc/CodeTracer`.

## 1. Introduction

The unprecedented capabilities of large language models (LLMs) in code generation have introduced critical challenges for intellectual property protection and code attribution (Li et al., 2022; Achiam et al., 2023; Guo et al., 2024). As AI systems produce increasingly sophisticated code that is difficult to distinguish from human-written code, the need for reliable code tracing approaches has become urgent (Zhao et al., 2024; Wang et al., 2024). Traditional code watermarking approaches apply post-generation transformations to completed code, making them inherently incompatible with the autoregressive generation process of LLMs. Moreover, these methods require labor-intensive, language-specific transformation rules that must be manually crafted for each programming language (Hamilton & Danicic, 2011; Yang et al., 2023; Li et al., 2025).

Existing LLM watermarking approaches operate on the autoregressive generation process by biasing next-token predictions toward statistically detectable patterns (Radford et al., 2019; Kirchenbauer et al., 2023a). In natural language contexts, this approach succeeds because text generation is robust, as most positions admit multiple semantically valid token choices, allowing watermarking to be applied across all generated tokens (Kuditipudi et al., 2023; Dathathri et al., 2024; Liu & Bu, 2024; Zhao et al., 2024).

However, code watermarking presents distinct challenges that arise from the structural and semantic constraints inherent to programming languages. Code generation imposes two critical constraints. First, syntactic dependencies severely constrain the space of valid token choices, as certain tokens are syntactically mandatory and their modification results in compilation failures (Guan et al., 2024). Second, code positions exhibit heterogeneous sensitivity to modifications; indiscriminate watermarking strategies fail to account for the varying tolerance to perturbations across different code locations (Lee et al., 2023). Recent efforts to incorporate watermarks during LLM code generation show promise but remain impractical in real-world scenarios, as they require access to supplementary prompts and model information to compute critical values like entropy during detection (Lee et al., 2023; Guan et al., 2024; Zhao et al., 2024). We argue that an effective solution requires a model-based approach that learns the necessary programming knowledge during training, enabling intelligent watermarking decisions that adapt to syntactic and semantic constraints with minimal context.

Consequently, we identify the core challenge in LLM code watermarking: *how can we intelligently determine optimal watermark insertion points and select semantically reasonable token choices that maintain statistical detectability while preserving code functionality?*

In this work, we propose CodeTracer, an adaptive code wa-

[1]Penn State University, USA [2]University of Washington, USA [3]Allen Institute for AI (AI2), USA. Correspondence to: Zhimeng Guo <zzg5107@psu.edu>.

*Proceedings of the 43rd International Conference on Machine Learning*, Seoul, South Korea. PMLR 306, 2026. Copyright 2026 by the author(s).

termarking framework built upon a novel reinforcement learning (RL) training paradigm. CodeTracer employs a policy-driven approach that leverages a parameterized model to intelligently identify optimal insertion positions and guide token selection during the code generation process. The parameterized model collaborates with the original LLM to form a watermarked policy, where only the parameterized model undergoes optimization during training. At generation time, the LLM provides logits while the parameterized model intelligently determines watermark application and token selection decisions. To train the watermarked policy, we employ reinforcement learning through a carefully designed dual-component reward system. This system integrates two complementary feedback signals: execution feedback that penalizes functionally incorrect code, and watermark embedding signals that comprise both immediate process rewards for successful watermarked token selection and statistical outcome rewards employing metrics such as the z-score to comprehensively assess detectability performance. To enable end-to-end gradient-based optimization, we address the non-differentiability of discrete bias token selection via Gumbel Top-k reparameterization (Xie & Ermon, 2019) and Straight-Through Estimation (Bengio et al., 2013), achieving differentiable policy training.

**Contributions.** Our work yields several key contributions: (i) We introduce CodeTracer, an adaptive watermarking framework that intelligently embeds watermarks during LLM code generation; (ii) We develop a novel RL pipeline for code watermarking training that combines execution feedback with dual watermark signals and enables differentiable optimization for the entire pipeline; and (iii) Empirically, we validate the effectiveness of CodeTracer through comprehensive evaluations, demonstrating its watermark detection capabilities while maintaining code functionality.

## 2. Related Work

**LLM Watermarking.** Existing LLM watermarking approaches embed imperceptible signatures during token sampling by modifying logits or altering the sampling procedure (Kirchenbauer et al., 2023a; Kuditipudi et al., 2023; Zhao et al., 2023; Christ et al., 2024; Dathathri et al., 2024). A prominent example is the green-red watermarking scheme (Kirchenbauer et al., 2023a;b), which partitions the vocabulary into "green" (preferred) and "red" (avoided) tokens, biasing generation toward green tokens while suppressing red ones, enabling statistical detection of watermarked content. Xu et al. (2024) propose a reinforcement learning approach for watermark embedding that requires training the LLM, which may cause unexpected behaviors for LLMs. Critically, these methods encounter difficulties in low-entropy scenarios typical of code generation.

**Code Watermarking.** Code watermarking presents dis-

tinct challenges due to the strict syntactic and semantic constraints inherent in programming languages. Traditional approaches modify existing code through formatting changes or control flow restructuring (Hamilton & Danicic, 2011; Ma et al., 2019; Li et al., 2025; Liu et al., 2024; Dathathri et al., 2024; Yang et al., 2023; Lin et al., 2026). Recent LLM-focused methods leverage entropy distributions (Lee et al., 2023; Li et al., 2023) or type predictors (Guan et al., 2024) to guide watermark insertion. However, these techniques typically require privileged access to LLM parameters, generation probabilities, or original prompts during watermark detection, significantly limiting their practical deployment (Zhao et al., 2024). In contrast, CodeTracer adaptively embeds watermarks during generation without such prerequisites.

**RLVR and GRPO.** Reinforcement Learning with Verifiable Rewards (RLVR) enhances LLMs by integrating verifiable reward signals into the training process, demonstrating improved robustness against reward hacking and superior performance (Lambert et al., 2024; Guo et al., 2025; Team et al., 2025). However, RLVR is constrained by the scarcity of reliably verifiable signals, with most applications limited to mathematical problems and code execution tasks. DeepSeek-R1 (Guo et al., 2025) combines verifiable rewards with Group Relative Policy Optimization (GRPO), which improves computational efficiency over traditional policy optimization methods. Code watermarking emerges as a natural fit for RLVR, as watermark detection provides unambiguous, token-level verification signals that can be efficiently computed. We exploit this alignment by employing GRPO with watermark-based verifiable rewards to achieve both computational efficiency and robust watermarking.

## 3. CodeTracer: A Policy-Driven Watermarking Framework

**Problem Setting.** In code generation tasks, a large language model takes a sequence of input tokens representing the prompt $\mathbf{x} = [x_1, x_2, \ldots, x_n]$ and generates an output sequence $\mathbf{y} = [y_1, y_2, \ldots, y_m]$ containing the generated code. At each generation step $t$, the LLM $\pi_\theta$ processes the input prompt $\mathbf{x}$ and previously generated tokens to compute a logit vector $\mathbf{l} = [l_1, l_2, \ldots, l_{|\mathcal{V}|}]$ over the vocabulary $\mathcal{V}$. Each value $l_j$ represents the model's preference for token $v_j \in \mathcal{V}$. This logit vector is transformed into a probability distribution using the softmax function, from which token $y_t$ is sampled. To enable detection of LLM-generated code, in-generation watermarking techniques, such as the approach by Kirchenbauer et al. (2023a), modify the LLM's original logits $\mathbf{l}$ to form watermarked logits $\tilde{\mathbf{l}}$. This modification is achieved by adding biases to the logits of a subset (green list) $G \subset \mathcal{V}$ of the vocabulary, selected using a pseudorandom function (PRF). This yields a watermarked code sequence

$\tilde{\mathbf{y}} = [\tilde{y}_1, \tilde{y}_2, \ldots, \tilde{y}_{m'}]$. The presence of a watermark is then statistically inferred by analyzing the frequency difference of tokens appearing within the biased vocabulary subset.

**Challenges in LLM Code Watermarking.** LLM watermarking faces severe performance degradation in code generation due to unique challenges distinct from natural text watermarking. Unlike natural language, code is highly structured with precise syntax where even small modifications can drastically alter functionality or render the code inoperable. This inherent rigidity creates significant constraints for watermarking techniques. First, watermark position selection is critical, as many positions in code are immutable. Unlike natural text where alterations rarely affect meaning, code contains structural elements that cannot be modified. For example, changing `def` to `func` in Python function definitions immediately breaks syntax. Any watermarking approach must avoid these critical positions. Second, watermark token choice must respect contextual constraints. Even at modifiable positions, replacement tokens must maintain syntactic validity. For instance, if `status = ''active''` is watermarked by replacing `''active''` with `class`, the result causes a syntax error. This contextual sensitivity severely limits the available vocabulary for watermarking. Consequently, effective code watermarking requires a deeper understanding of code structure and semantics.

### 3.1. CodeTracer

To address the aforementioned challenges, CodeTracer introduces a policy-driven framework that integrates a watermark model $\pi_\phi$ with an LLM $\pi_\theta$, yielding a composite watermarked policy $\pi_{\theta \oplus \phi}$ capable of generating watermarked code. At each generation step $t$, the watermark model $\pi_\phi(a|\mathbf{c})$ operates conditioned on the current context $\mathbf{c}$, defined as the concatenation of a segment of the input $\mathbf{x}$ and the sequence of previously generated tokens $\mathbf{y}_{<t}$ within a fixed-length window. The output of this policy is an action $a = (w, G)$, where $w \in \{0, 1\}$ is a binary variable indicating whether to apply watermarking at the current position, and $G \subset \mathcal{V}$ represents a set of preferred "green" tokens.

The generation of a token is then achieved by sampling from a modified logit vector from the watermarked policy $\pi_{\theta \oplus \phi}$ as:

$$\tilde{l}_j = l_j + w \cdot \delta \cdot \mathbb{1}_{v_j \in G}, \qquad (1)$$

where $\delta$ is a hyperparameter controlling the bias applied to the logits of tokens in the green list $G$ when watermarking is active ($w = 1$), and $\mathbb{1}_{v_j \in G}$ is an indicator function equal to 1 if token $v_j \in G$ and 0 otherwise. The watermark strength is jointly governed by the size of the green token set, $|G| = \gamma|\mathcal{V}|$ for a predefined ratio $\gamma \in (0, 1)$, and the bias magnitude $\delta$. The complement of the green set, $R = \mathcal{V} \setminus G$, constitutes the corresponding "red" token set.

In essence, this formulation empowers the policy to strategically influence the token sampling process by preferentially selecting tokens from the green set $G$, while simultaneously regulating the watermark through the binary decision $w$.

Consequently, during generation, CodeTracer seamlessly applies the watermarked policy $\pi_{\theta \oplus \phi}(\tilde{\mathbf{y}}|\mathbf{x})$ that takes the same input as the LLM $\pi_\theta$ but produces watermarked outputs $\tilde{\mathbf{y}} = [\tilde{y}_1, \tilde{y}_2, \ldots]$ via the modified logits $\tilde{l}_j$.

**Detection.** During detection, given an output sequence $\mathbf{s} = [s_1, s_2, \ldots, s_{T'}]$ of length $T'$, CodeTracer reconstructs the watermarking decisions independently using only the watermark model $\pi_\phi$, without requiring access to the LLM $\pi_\theta$. This reconstruction determines, for each token, whether watermarking was applied ($w = 1$ or $w = 0$) and, if so, the composition of the corresponding green token set $G$. We identify the subset of tokens where watermarking was active, denoted as $\{s_{w=1}\}$, and count the total number of watermarked positions $T = |\{s_{w=1}\}|$. We then count how many watermarked tokens belong to their respective reconstructed green sets, denoted as $N_G = |\{s : s \in \{s_{w=1}\} \wedge s \in G\}|$. To assess the statistical significance, we employ a one-proportion $z$-test for the proportion of watermarked tokens belonging to their predicted green sets:

$$z = \frac{N_G - T\gamma}{\sqrt{T\gamma(1 - \gamma)}}. \qquad (2)$$

Under the standard null hypothesis of no watermarking, we expect $N_G$ to follow a binomial distribution with success probability $\gamma$. A sufficiently large positive $z$-score then provides strong statistical evidence for watermark presence, indicating that watermarked tokens appear in their predicted green sets with frequency significantly higher than the expected random chance baseline of $\gamma$. The complete algorithms for watermark generation and detection are formalized in Algorithm 1 and Algorithm 2, respectively.

## 4. Learning to Watermark in CodeTracer

We formulate training of watermarked policy $\pi_{\theta \oplus \phi}$ within a reinforcement learning framework. This section first gives an overview of the training pipeline, then discusses the reason to use RL for policy learning, and finally addresses specific design challenges along with our proposed approach.

### 4.1. How to Learn the Watermarked Policy?

As shown in Figure 1, we employ the GRPO algorithm (Shao et al., 2024) as our RL framework to train the watermark model $\pi_\phi(a|\mathbf{c})$ within the watermarked policy $\pi_{\theta \oplus \phi}$. GRPO offers notable computational efficiency and has demonstrated strong effectiveness in integrating rule-based rewards for coding and mathematical tasks (Guo et al., 2025). Our training pipeline requires three key com-

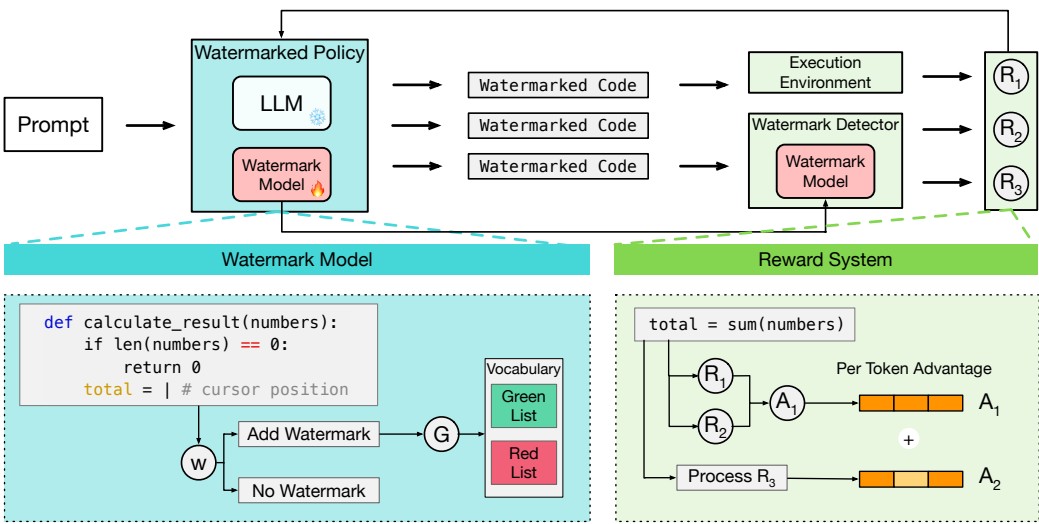

*Figure 1.* **CodeTracer**: A framework for LLM code watermarking through selective token biasing. The diagram shows our end-to-end pipeline where a trainable watermark model collaborates with an LLM to embed detectable statistical patterns in generated code. A reward system optimizes the dual objectives of preserving code functionality while maximizing watermark detectability. The watermark model operates as a plug-in module, enabling deployment beyond those used during training.

ponents from the standard GRPO framework: a training policy, a reference policy, and reward functions. We use the integrated watermarked policy $\pi_{\theta \oplus \phi}$ as the training policy rather than solely $\pi_\phi$, as the complete code generation process requires coordinated operation of both the LLM and watermark model. Crucially, we entirely freeze the LLM parameters $\theta$ during training and optimize only the watermark model parameters $\phi$. This design ensures a portable and efficient watermarking solution such that the trained watermark model can be applied to other pre-trained LLMs without requiring fine-tuning or additional training data.

### 4.2. Why Use RL for Learning?

Code watermarking presents a fundamental training paradox that reinforcement learning uniquely addresses. Traditional supervised learning requires pre-existing watermarked examples to train a detector, but creating such examples necessitates an already-trained watermark model: a circular dependency that renders supervised approaches infeasible. Reinforcement learning eliminates this paradox by learning through interaction rather than predefined examples.

Code watermarking is naturally suited for RL due to two key characteristics. First, watermark detection provides unambiguous, token-level verification signals that can be efficiently computed. Second, code functionality offers clear binary feedback through test case execution. Our RL formulation leverages these signals to offer distinct advantages: (1) it implicitly learns complex syntactic constraints across diverse programming languages without exhaustive manual specification; (2) it optimizes the watermarked policy to balance detectability and functionality through quantifi-

able reward mechanisms; (3) it enables end-to-end training without requiring pre-watermarked data.

### 4.3. Policy Design

The implementation of our watermarked policy $\pi_{\theta \oplus \phi}$ presents fundamental technical challenges due to the discrete nature of the watermark components $w$ and $G$ in Equation 1. We implement the watermark model $\pi_\phi(a|\mathbf{c})$ where $a = (w, G)$ as a transformer that outputs continuous values, but we need discrete outputs for watermarking decisions. Both the binary watermark decision $w$ and the green token list $G$ require discrete selections that impede gradient flow during training.

**Straight-Through Estimation.** To enable gradient flow through the discrete decision processes for watermark selection $w$, we apply straight-through estimation techniques. Specifically, our watermark transformer outputs a $(|\mathcal{V}| + 1)$-dimensional vector as $(w_\phi, l_\phi)$, where $w_\phi \in \mathbb{R}$ is a scalar for watermark placement probability, and $l_\phi \in \mathbb{R}^{|\mathcal{V}|}$ is a logit bias vector. To derive our desired output $a = (w, G)$ from $(w_\phi, l_\phi)$, we implement straight-through estimation:

$$w = \mathbb{1}_{w_\phi > 0} + \sigma(w_\phi) - \text{sg}(\sigma(w_\phi)), \qquad (3)$$

where $\sigma(\cdot)$ is the standard sigmoid function and $\text{sg}(\cdot)$ denotes the stop-gradient operator. During the forward pass, this formulation behaves exactly like the discrete operation $\mathbb{1}_{w_\phi > 0}$, producing deterministic hard binary decisions. However, during backpropagation, gradients seamlessly flow through the continuous relaxation $\sigma(w_\phi)$. The stop-gradient operator crucially ensures that only the gradient

of the relaxation (and not its forward values) affects the overall computation. This technique enables robust end-to-end training of the watermark model despite the inherently discrete nature of watermarking decisions governed by $w$.

**Gumbel-Top-k.** To overcome the challenges of discrete token selection for the green list $G$, we develop a Gumbel-Top-k sampling approach for our token assignment process. This technique provides a differentiable pathway from $l_\phi$ to $G$ while preserving the discrete characteristics necessary for effective watermarking. For a given parameter $\gamma \in (0, 1)$, we select $k = \lfloor \gamma |\mathcal{V}| \rfloor$ tokens for the green list. Given logits $\mathbf{l}_\phi$ from the watermark model $\pi_\phi$, we apply Gumbel noise to create a continuous relaxation of the discrete selection:

$$\mathbf{g} = \mathbf{l}_\phi + (-\log(-\log(\mathbf{u}))), \qquad (4)$$

where $\mathbf{u} \sim \text{Uniform}(0, 1)^{|\mathcal{V}|}$ represents uniform random noise across the vocabulary space. The green list $G$ is then determined by selecting the top-$k$ tokens according to these perturbed logits $\mathbf{g}$:

$$G = \arg \text{top-}k(\mathbf{g}). \qquad (5)$$

To enable gradient flow through this discrete selection process, we implement a straight-through estimator for the indicator variable $\mathbf{l}_G \in \{0, 1\}^{|\mathcal{V}|}$ that represents membership in the green list:

$$\mathbf{l}_G = \mathbb{1}_{v \in G} + \mathcal{S}(\mathbf{g}) - \text{sg}(\mathcal{S}(\mathbf{g})), \qquad (6)$$

where $\mathcal{S}(\mathbf{g})$ represents the Gumbel-Softmax-based continuous relaxation of the top-$k$ selection, and $\text{sg}(\cdot)$ is the stop-gradient operator. During forward passes, we use hard discrete token selection $\mathbb{1}_{v \in G}$ with Equation 5 based on token rankings from the Gumbel-perturbed logits $\mathbf{g}$, while during backward passes, the gradients smoothly flow through the softmax-based continuous relaxation $\mathcal{S}(\mathbf{g})$. This approach allows the model to learn effective and robust watermarking while maintaining the required discrete token selection.

For the reference policy in our GRPO framework, we utilize the exact same architecture as the training policy $\pi_{\theta \oplus \phi}$, establishing a self-referential optimization approach. This design choice enables the watermarking parameters to be refined against their own previous distributions. The approach facilitates gradual refinement of the watermarked policy while ensuring that successive iterations remain coherent with previous behaviors, ultimately leading to more stable convergence and improved overall performance.

### 4.4. Reward Function Design

Our reward system guides the watermarked policy $\pi_{\theta \oplus \phi}$ in watermark placement $w$ and green token selection $G$ using feedback from watermark detection and code execution. We design a multifaceted reward system that balances code functionality and watermark detectability.

**Outcome-Based Execution Reward $R_1$.** Our execution reward $R_1$ evaluates code functional correctness through binary assessment based on passing test cases:

$$R_1 = \begin{cases} 1, & \text{if all test cases pass} \\ 0, & \text{otherwise} \end{cases} \qquad (7)$$

This binary formulation provides unambiguous feedback on code correctness, where successful execution across all test cases yields a positive reward, while any compilation error, runtime failure, or test failure results in zero reward. By incorporating this strict binary signal into our reward function, we enforce functional preservation as a hard constraint, ensuring that the watermarked policy maintains code correctness while learning watermarking strategies.

**Outcome-Based Watermark Reward $R_2$.** The watermark detection reward $R_2$ quantifies the statistical detectability of the watermark by employing a saturated function of the statistical $z$-score:

$$R_2(s) = \begin{cases} 1, & \text{if } z(s) \geq 4 \\ \frac{z(s)}{4}, & \text{if } 0 < z(s) < 4 \\ 0, & \text{if } z(s) \leq 0 \end{cases} \qquad (8)$$

where $z(s)$ is the statistical $z$-score obtained from the detection algorithm as a holistic measurement of watermark performance. The saturation threshold of 4 is not an arbitrary heuristic but is derived directly from the one-proportion $z$-test statistic used to detect watermarks (Kirchenbauer et al., 2023a): a $z$-score of 4 corresponds to a $p$-value of approximately $3 \times 10^{-5}$, representing a standard threshold for high statistical significance in hypothesis testing. This saturated reward formulation establishes clear boundaries: a $z$-score of 4 or higher receives the maximum reward of 1, while non-positive $z$-scores receive no reward. Between these thresholds, the reward scales linearly with the $z$-score. This design incentivizes our policy to achieve statistically significant detection levels while preventing extreme $z$-scores from skewing the advantage computation.

**Process-Based Watermark Reward $R_3$.** We observed that relying solely on outcome-based rewards can be insufficient for effective policy learning, as they provide identical rewards for tokens in a sequence. To provide more granular guidance, we introduce a process-based reward $R_3$ that evaluates individual generated tokens, offering immediate feedback on token selection decisions. This reward is straightforward to implement, as we can directly assess whether each generated token $s_t$ is green, red, or non-watermarked

based on the policy's action $a_t = (w_t, G_t)$ at time $t$:

$$R_3(s_t, a_t) = \begin{cases} 1, & \text{if } w_t = 1 \text{ and } s_t \in G_t \\ -1, & \text{if } w_t = 1 \text{ and } s_t \in R_t \\ 0, & \text{if } w_t = 0 \end{cases} \quad (9)$$

This reward structure encourages green token selection during watermarking while penalizing red token selection. Non-watermarked positions receive neutral rewards, concentrating the learning signal on watermarked tokens. By providing immediate feedback at each generation step, $R_3$ delivers more granular guidance than sequence-level rewards alone, accelerating policy convergence and improving watermarking performance through token-level supervision.

**Integration into Advantage Computation.** Our approach calculates advantages at two complementary levels to optimize the watermarked policy. First, we compute outcome advantages $A_1$ from the execution reward $R_1$ and watermark detection reward $R_2$ using GRPO's outcome-based group normalization mechanism. Second, we derive process advantages $A_2$ from the token-level watermark reward $R_3$, normalizing these values across all tokens in all rollouts for each prompt. We categorize generated tokens into four distinct types: green tokens (watermarked and belonging to set $G$), red tokens (watermarked but falling in set $R$), non-watermarked tokens (where $w = 0$), and non-code text tokens. To maintain a specific focus on code watermarking, we exclude non-code tokens entirely.

The final advantage function for each token $s_t$ at time $t$ integrates these signals directly:

$$A_{\text{total}}(s_t, a_t) = A_1 + A_2(s_t, a_t), \quad (10)$$

where $A_1$ represents outcome-based advantage and $A_2$ represents process-based advantage for token $s_t$ with action $a_t$. We then apply a binary mask to eliminate advantage values for non-code tokens:

$$\hat{A}(s_t, a_t) = A_{\text{total}}(s_t, a_t) \cdot \mathbb{1}_{\text{is\_code}}(s_t). \quad (11)$$

This advantage function balances sequence-level feedback with token-specific guidance. $A_1$ provides sequence-level guidance from watermark effectiveness and functional correctness, while $A_2$ delivers token-level feedback for watermark positioning and green token selection.

Intuitively, our carefully integrated reward system assigns high advantages to tokens that simultaneously maintain code functionality, contribute to effective watermark detection, and are selected from the green set. This approach guides our watermarked policy to identify optimal watermarking positions and select tokens that preserve functionality while introducing subtle, statistically detectable deviations.

**KL Regularization.** We include a Kullback-Leibler (KL) divergence penalty in the general GRPO objective to regularize the watermarked policy $\pi_{\theta \oplus \phi}$ against the reference policy $\pi_{\text{ref}}$ when stronger distributional control is desired. This term, $D_{\text{KL}}(\pi_{\theta \oplus \phi} \parallel \pi_{\text{ref}})$, is controlled by the coefficient $\beta$; in our reported experiments, hyperparameter tuning selected $\beta = 0$, so the KL term is disabled while the reward and clipping terms provide the effective constraints.

**Objective of GRPO.** Combining all components discussed above, the final GRPO objective for our watermarked policy can be concisely expressed as:

$$\max_{\phi} \mathbb{E}_{s \sim \mathcal{D}} \left[ \frac{1}{|s|} \sum_{t=1}^{|s|} \min \left[ \frac{\pi_{\theta \oplus \phi}(s_t | s_{<t})}{\pi_{\text{ref}}(s_t | s_{<t})} \hat{A}(s_t, a_t), \right.\right.$$
$$\left.\left. \text{clip} \left( \frac{\pi_{\theta \oplus \phi}(s_t | s_{<t})}{\pi_{\text{ref}}(s_t | s_{<t})}, 1 - \varepsilon, 1 + \varepsilon \right) \hat{A}(s_t, a_t) \right] \right]$$
$$- \beta D_{\text{KL}}(\pi_{\theta \oplus \phi} \parallel \pi_{\text{ref}}), \quad (12)$$

where $\mathcal{D}$ represents rollout data from the watermarked policy, $\hat{A}(s_t, a_t)$ is our masked advantage function incorporating both outcome-based and process-based rewards, $\varepsilon$ is a clipping parameter that limits policy updates, and $\beta$ controls KL regularization. This objective balances watermark detectability, code functionality, and natural generation patterns while only optimizing watermark parameters $\phi$.

## 5. Experiments

**Evaluation Overview.** We evaluate `CodeTracer` across five critical dimensions: (i) code functionality, (ii) watermark detectability, (iii) robustness against attacks, (iv) efficiency and (v) transferability. Following Lee et al. (2023), we employ two primary metrics: Pass@k for functionality assessment and AUROC and TPR@5%FPR for detection performance measurement (Chen et al., 2021).

**Models and Datasets.** `OpenCoder-1.5B-Instruct` (Huang et al., 2024) is our primary backbone LLM. To validate transferability, we train on a 1.5B model and directly evaluate on `OpenCoder-8B-Instruct` without retraining. We evaluate performance on three established benchmarks: `HumanEval` (Chen et al., 2021) and `MBPP` (Austin et al., 2021) for Python evaluation, and `HumanEvalPack` (Muennighoff et al., 2023) for cross-language evaluation.

**Baselines.** We compare `CodeTracer` against two categories of baselines. For post-hoc detection methods, we evaluate against `logp(x)`, `LogRank` (Gehrmann et al., 2019), `DetectGPT` (Mitchell et al., 2023), and `GPTZero` (Tian & Cui, 2023). For active watermarking approaches, we compare with `WLLM` (Kirchenbauer et al., 2023a), `EXP-edit` (Kuditipudi et al., 2023) and `SWEET` (Lee et al., 2023).

**Technical Implementation.** Detailed implementation spec-

*Table 1.* Performance (%) comparison of detection methods. **Bold**: best performance for watermarks.

| Category | Dataset | HumanEval | | | | MBPP | | | |
|---|---|---|---|---|---|---|---|---|---|
| | Method | Pass@1 | Pass@10 | AUROC | TPR | Pass@1 | Pass@10 | AUROC | TPR |
| *No Watermark* | Base | 65.42 | 79.17 | - | - | 43.35 | 51.65 | - | - |
| *Post-hoc* | logp(x) | 65.42 | 79.17 | 47.59 | 4.27 | 43.35 | 51.65 | 47.77 | 6.40 |
| | LogRank | 65.42 | 79.17 | 47.66 | 1.82 | 43.35 | 51.65 | 48.76 | 7.80 |
| | DetectGPT | 65.42 | 79.17 | 51.12 | 9.15 | 43.35 | 51.65 | 46.15 | 3.60 |
| | GPTZero | 65.42 | 79.17 | 52.00 | 5.50 | 43.35 | 51.65 | 41.10 | 2.80 |
| *Watermarks* | WLLM | 58.05 | 70.35 | 70.17 | 20.73 | 39.66 | 47.22 | 76.44 | 27.80 |
| | EXP-edit | 59.29 | 72.41 | 66.50 | 25.61 | 40.16 | 50.25 | 51.22 | 10.60 |
| | SWEET† | 60.46 | 74.11 | 76.24 | 27.44 | 39.64 | 47.47 | 77.24 | 24.80 |
| | SWEET‡ | 61.65 | 76.73 | 71.19 | 17.07 | 42.06 | 49.22 | 76.57 | 28.40 |
| | CodeTracer | **62.65** | **77.11** | **77.71** | **32.32** | **42.10** | **52.01** | **78.42** | **31.60** |

SWEET† and SWEET‡ use different entropy thresholds to explore detectability-functionality trade-offs. We adapt SWEET to use fixed context windows (vs. full generation history and original LLM) for practical and fair comparison with other baselines. See Appendix C.2 for details about entropy threshold setup.

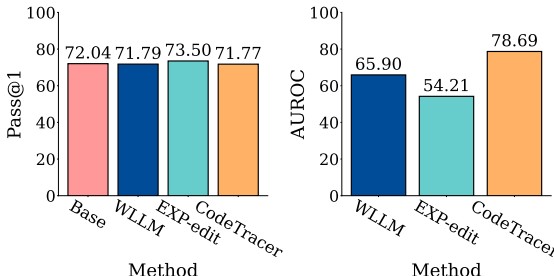

*Figure 2.* Training on 1.5B LLM and evaluating on 8B `OpenCoder-8B-Instruct` LLM.

*Table 2.* Relative computational overhead of `CodeTracer` with a 1.5B base LLM. Our watermark model runs in parallel, so the measured serial delay is only the tensor-slicing delay.

| Component | Parameters | Memory | Delay |
|---|---|---|---|
| CodeTracer $\pi_\phi$ | 118M | < 0.5GB | < 100 $\mu s$ |
| 1.5B LLM $\pi_\theta$ | 1,500M | 3-6GB | 500-800ms |
| **Relative Overhead** | **< 10%** | **< 15%** | **< 0.02%** |

ifications, including dataset details, baseline configurations, model architectures, and training details, are provided in Appendix C. We mainly follow the experiment settings of Lee et al. (2023) and adapt them to our setting. In accordance with established reinforcement learning practices, we initially apply supervised fine-tuning to $\pi_\phi$ using proxy signals to establish code distribution knowledge through next-token prediction. The training process only needs to be performed once and can be completed in 1 day on a single A100 GPU. We can use the trained watermark model to watermark other pre-trained LLMs without requiring fine-tuning.

### 5.1. Main Results on Benchmark Datasets

We present experiments evaluating `CodeTracer`'s performance on code functionality and watermark detection with `HumanEval` and `MBPP` in Table 1. We observe a clear improvement over other baselines.

**Post-hoc vs. Watermarks.** Post-hoc detection methods, including `logp(x)`, `LogRank`, `DetectGPT`, and `GPTZero`, preserve original generated code but demonstrate consistently poor discrimination performance. Their AUROC values range from 47.59% to 52.00% on `HumanEval` and 41.10% to 48.76% on `MBPP`, close to random guessing (50%). This indicates that with the advancement of LLMs,

post-hoc detection methods struggle to distinguish AI-generated code from human-written code.

**Code Functionality and Watermark Detectability.** Watermarking methods demonstrate superior detection performance compared to post-hoc approaches, achieving AUROC values of 66.50-77.71% versus 47.59-52.00%. However, existing watermarking baselines suffer from noticeable code quality degradation, with Pass@1 scores dropping to 58.05-61.65% on `HumanEval`, representing a substantial trade-off between detection capability and functionality. `CodeTracer` achieves considerably better balance in this trade-off with the highest Pass@1 scores (62.65% on `HumanEval`, 42.10% on `MBPP`) and superior detection capability (AUROC of 77.71% and 78.42%), consistently outperforming SWEET and other baselines across both functionality and watermark detectability metrics.

**Computation Efficiency.** As shown in Table 2, `CodeTracer` introduces negligible computational overhead with less than 10% parameter increase, minimal memory usage (<15%), and inference delay below 0.02%. The watermark model adds only 118M parameters while requiring minimal additional memory (less than 0.5GB). Because the watermark model runs in parallel with the base LLM, the additional serial overhead is dominated by efficient tensor slicing operations, introducing less than 100 $\mu s$—orders of

*Table 3.* Robustness evaluation (%) under code modification attacks. **Bold** values indicate best performance. Settings follow Lee et al. (2023).

| Attack | Metric | WLLM | EXP-edit | SWEET | CodeTracer |
|---|---|---|---|---|---|
| *Original* | AUROC | 70.17 | 66.50 | 76.24 | **82.95** |
| | TPR | 20.73 | 25.61 | 27.44 | **46.34** |
| *DIPPER* | AUROC | 55.92 | 51.21 | 53.05 | **58.42** |
| | TPR | 12.81 | 8.54 | 11.39 | **14.31** |
| *Rename* | AUROC | 70.91 | 62.02 | 68.65 | **73.36** |
| | TPR | 20.12 | 9.76 | 14.02 | **29.11** |

*Table 4.* Ablation study on different reward components for `CodeTracer` training.

| Method | Pass@1 (%) | AUROC (%) | TPR (%) |
|---|---|---|---|
| CodeTracer | 60.82 | 82.95 | 46.34 |
| w/o $A_2$ | 61.15 (+0.33) | 75.11 (-7.84) | 30.29 (-16.05) |
| w/o $A_1$ | 60.34 (-0.48) | 79.52 (-3.43) | 34.91 (-11.43) |

*Table 5.* Performance of model variants.

| Method | Pass@1 (%) | AUROC (%) | TPR (%) |
|---|---|---|---|
| CodeTracer-1 | 62.65 | 77.71 | 32.32 |
| CodeTracer-2 | 60.82 | 82.95 | 46.34 |

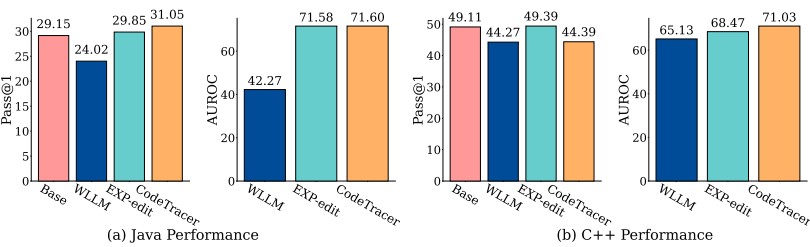

(a) Java Performance    (b) C++ Performance

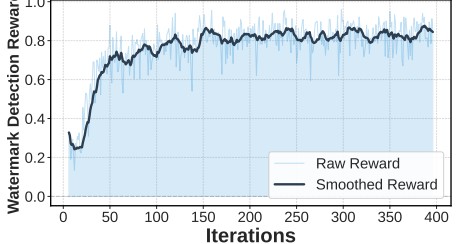

*Figure 4.* Training curve for `CodeTracer-1`.

*Figure 3.* Cross-language evaluation. `CodeTracer` shows non-trivial zero-shot transfer within the same model/tokenizer family.

magnitude smaller than typical LLM latency. This minimal overhead makes `CodeTracer` practical for real-world deployment without compromising user experience.

**Transferability to Larger Models.** To verify the scalability of our approach, we train the watermark model only once on a smaller 1.5B LLM and directly apply it as a lightweight plug-in module to the larger `OpenCoder-8B-Instruct` without any retraining. Results are shown in Figure 2. `CodeTracer` achieves minimal code functionality degradation (71.77% Pass@1 vs. 72.04% for Base) while significantly outperforming other watermarking methods in detection capability (78.69% AUROC compared to 65.90% for `WLLM` and 54.21% for `EXP-edit`). This demonstrates that our watermark model can serve as an effective plug-in module for larger models without requiring retraining.

**Robustness Analysis.** Table 3 evaluates `CodeTracer`'s robustness against DIPPER (Krishna et al., 2024) paraphrasing and variable renaming attacks (Lee et al., 2023). While all methods degrade under attacks, `CodeTracer` consistently outperforms baselines. Against DIPPER attacks, `CodeTracer` maintains superior detection (AUROC 58.42%, TPR 14.31%), and under renaming attacks demonstrates resilience (AUROC 73.36%, TPR 29.11%), indicating effectiveness in real-world scenarios.

**Cross-Language Capabilities.** Figure 3 shows `CodeTracer`'s cross-language performance on Java and C++. Since the watermark policy is trained mainly on Python code, these results should be interpreted as non-trivial zero-shot transfer within the same model/tokenizer family rather than language-agnostic

generalization. The comparison with SWEET is mixed but competitive: on Java, SWEET achieves 25.22% Pass@1 and 70.31% AUROC, while `CodeTracer` achieves 31.05% Pass@1 and 71.60% AUROC; on C++, SWEET achieves 44.85% Pass@1 and 70.09% AUROC, while `CodeTracer` achieves 44.39% Pass@1 and 71.03% AUROC.

## 5.2. Further Analysis

**Reward Components.** Table 4 evaluates the impact of different reward components on `CodeTracer`'s performance. `CodeTracer` achieves optimal performance with both process and outcome rewards. Ablating either component leads to notable performance degradation, with removing the process reward causing larger detection drops while removing the outcome reward affects both functionality and detection.

**Performance Tradeoff.** Table 5 demonstrates the controllable trade-off between code functionality and watermark detectability. `CodeTracer-1` uses solely RL training without supervised fine-tuning initialization, achieving higher functionality (Pass@1 62.65%) but lower detection performance (AUROC 77.71%). Figure 4 shows the training dynamics of `CodeTracer-1`, demonstrating consistent improvement throughout RL training. In contrast, `CodeTracer-2` incorporates both supervised fine-tuning and RL, trading some functionality for significantly improved detection capability.

## 6. Conclusion

We present CodeTracer, a novel adaptive code watermarking framework that addresses the challenges of watermarking LLM-generated code while preserving functionality. Our policy-driven approach integrates a lightweight watermark model with the base LLM to form a composite policy that dynamically decides when to watermark and which tokens to select. By leveraging reinforcement learning through GRPO, our approach intelligently navigates programming language constraints without requiring pre-existing watermarked examples. Our multi-component reward system balances statistical detectability with code integrity through execution feedback and token-level guidance. Extensive evaluations demonstrate that CodeTracer significantly outperforms existing approaches across multiple dimensions. Our method achieves superior detection capability while maintaining the highest code functionality among baselines. Notably, CodeTracer introduces negligible computational overhead, making it practical for real-world deployment.

## 7. Limitations

CodeTracer has two main limitations. First, it is more expensive to set up than fixed-rule watermarks. Training the watermarking policy $\pi_\phi$ with reinforcement learning requires repeated generation and reward computation, so the cost grows with the model size and the number of programming languages we support. This is a one-time training cost, but it makes quickly adapting CodeTracer to a new model or language less convenient than methods that need no training. Second, our robustness is measured against the attacks we tested, such as code refactoring and variable renaming. Stronger or future attacks may still weaken the watermark, so keeping CodeTracer robust will require ongoing evaluation as new evasion techniques appear.

At inference time, the watermarked policy $\pi_{\theta \oplus \phi}$ chooses a new green/red token split at every step instead of reusing a fixed split, which adds a small overhead compared with simpler watermarks; in practice this cost is minor, but it scales with the size of the learned policy. Finally, we use Gumbel Soft Top-k to make the discrete green/red partition differentiable during training. This is an approximation of true discrete selection, and how well it behaves on rare or unusual code patterns is not yet fully understood. We leave a closer study of this approximation, and ways to reduce its inference overhead, to future work.

## Impact Statement

This work aims to advance responsible deployment of code-generating LLMs by enabling reliable identification of AI-generated code. As LLMs become increasingly capable of producing functional code, the ability to distinguish AI-generated content from human-written code becomes important for maintaining accountability, protecting intellectual property, and preserving academic integrity. While watermarking techniques could potentially be circumvented by determined adversaries, our approach raises the bar for such attempts while introducing minimal overhead. We believe that transparent attribution of AI-generated content contributes positively to the trustworthy integration of AI systems in software development workflows.

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

# A. Watermark Generation and Detection Algorithm of CodeTracer

## A.1. Watermark Generation

Algorithm 1 presents our watermark generation process. The algorithm embeds watermarks into code sequences while preserving their functionality. It takes as input a prompt $\mathbf{x}$, context window size $c$, and watermarking hyperparameters including green list ratio $\gamma$ and bias $\delta$ that control the strength and detectability of the embedded watermark.

At each timestep $t$, the watermark model processes the context window to determine whether to apply watermarking ($w$) and which tokens to include in the green list ($G$). The algorithm modifies logits selectively based on these decisions, applying bias $\delta$ to green list tokens when $w = 1$.

---

**Algorithm 1** CodeTracer Watermark Generation

---

**Require:** Prompt $\mathbf{x}$, Context window size $c$, Green list ratio $\gamma$, Bias strength $\delta$
**Ensure:** Watermarked code sequence $\tilde{\mathbf{s}}$
 1: Initialize empty sequence $\tilde{\mathbf{s}} = []$
 2: **for** each generation step $t$ **do**
 3:     Compute base LLM logits $\mathbf{l} = \pi_\theta(\mathbf{x}, \tilde{s}_{<t})$
 4:     Compute context window $\mathbf{ctx} = \text{concat}(\mathbf{x}, \tilde{\mathbf{s}})[-(c):]$
 5:     Compute watermark model outputs $(w_\phi, \mathbf{l}_\phi) = \pi_\phi(\mathbf{ctx})$
 6:     Compute watermark decision $w = \mathbb{1}_{w_\phi > 0}$
 7:     **if** $w = 1$ **then**
 8:         Select green list $G = \arg\text{top-}k(\mathbf{l}_\phi)$ and red list $R = \mathcal{V} \setminus G$, where $k = \lfloor \gamma |\mathcal{V}| \rfloor$
 9:         Compute modified logits: $\tilde{l}_j = l_j + \delta \cdot \mathbb{1}_{v_j \in G}$ for each token $v_j$ in vocabulary $\mathcal{V}$
10:     **else**
11:         Set $\tilde{\mathbf{l}} = \mathbf{l}$ (no modification)
12:     Sample next token from distribution: $\tilde{s}_t \sim \text{softmax}(\tilde{\mathbf{l}})$
13:     Append $\tilde{s}_t$ to $\tilde{\mathbf{s}}$
14: **return** $\tilde{\mathbf{s}}$

---

## A.2. Watermark Detection

Algorithm 2 details our statistical watermark detection procedure. Following the framework of Kirchenbauer et al. (2023a), we employ hypothesis testing with a key modification, where testing is performed selectively only at positions likely to contain watermarks.

Given a code sequence $\mathbf{s}$, the detector first reconstructs the vocabulary partitions and switch probabilities using the same watermark model from generation. Detection employs a one-sided hypothesis test using the z-statistic to determine whether green tokens appear more frequently than random chance would predict, with a positive z-score above the threshold indicating watermark presence.

# B. Advanced Training and Generation Techniques

## B.1. SFT Initialization

While our reinforcement learning (RL) algorithm can train from random initialization, we find that supervised fine-tuning (SFT) provides a significantly more effective starting point. Since watermarked code is unavailable for direct supervision, we instead use entropy and next-token prediction of clean code as a proxy. During SFT, the watermark model $\pi_\phi$ is trained on two complementary objectives: predicting watermark decisions $w_\phi$ to learn entropy distribution patterns across code structures, and optimizing logits $l_\phi$ to capture the underlying token distribution in code sequences. This dual-objective approach establishes a strong foundation that encodes both code syntax patterns and appropriate watermarking signals before RL optimization begins.

---

**Algorithm 2** CodeTracer Watermark Detection

---

**Require:** Code sequence $\mathbf{s}$, Context window size $c$, Green list ratio $\gamma$, Detection threshold $\tau$
**Ensure:** Detection result (watermarked/not watermarked/insufficient data)
 1: Initialize counters: watermarked_positions $= 0$, green_tokens $= 0$
 2: Set positions_checked $= \{\}$
 3: **for** each position $t$ in $\mathbf{s}$ **do**
 4:     Compute context window
      $\mathbf{ctx} = \mathbf{s}[\max(0, t - c) : t]$
 5:     Compute watermark model outputs $(w_\phi, \mathbf{l}_\phi) = \pi_\phi(\mathbf{ctx})$
 6:     Compute watermark decision $w = \mathbb{1}_{w_\phi > 0}$
 7:     **if** $w = 1$ **then**
 8:         Reconstruct green list $G = \arg$ top-$k(\mathbf{l}_\phi)$, where $k = \lfloor \gamma |\mathcal{V}| \rfloor$
 9:         watermarked_positions $++$
10:         **if** $s_t \in G$ **then**
11:            green_tokens $++$
12:         Add $t$ to positions_checked
13: Compute z-score:
    $z = \dfrac{\text{green\_tokens} - \text{watermarked\_positions} \cdot \gamma}{\sqrt{\text{watermarked\_positions} \cdot \gamma \cdot (1 - \gamma)}}$
14: **if** $z > \tau$ **then**
15:     **return** "watermarked"
16: **else**
17:     **return** "not watermarked"

---

## B.2. Improved Sampling Strategy

We implement a constrained sampling approach during both generation and detection phases to enhance watermark robustness without additional training. Rather than naively selecting the top-$k$ tokens by logit values, we perform a deterministic sorting of logits $l_\phi$ and then apply pairwise or tuple-based sampling constraints. This technique preserves the overall token distribution characteristics for unwatermarked positions while creating a more distinct statistical separation between "green" and "red" tokens for watermarked positions. The approach effectively maintains natural code generation quality while improving watermark detection reliability.

## B.3. Entropy Regularization

During policy optimization, we observed the watermark model occasionally converging toward degenerate solutions where watermark signals $w$ approach binary extremes (consistently 0 or 1). To mitigate this issue, we incorporate an entropy regularization term in the policy optimization objective:

$$L_{\text{entropy}} = -H(\pi_\phi(\sigma(w_\phi)|\mathbf{c})), \tag{13}$$

where $H(\pi_\phi)$ represents the entropy of the watermark decision distribution. This encourages the model to maintain more balanced probability distributions for watermark decisions, preventing overfitting and improving robustness across diverse code contexts. Additionally, we implement gradient clipping to further stabilize training dynamics.

## B.4. Reward Design for Trivial Solutions

Our initial reward design occasionally led to trivial solutions where the model would either apply watermarking too aggressively or too conservatively. To address this, we refined the process-based watermark reward $R_3$ with an asymmetric penalty structure:

$$R_3(s_t, a_t) = \begin{cases} 1, & \text{if } w_t = 1 \text{ and } s_t \in G_t \\ -\alpha, & \text{if } w_t = 1 \text{ and } s_t \in R_t \\ 0, & \text{if } w_t = 0 \text{ (non-watermarked)} \end{cases} \tag{14}$$

*Table 6.* Comprehensive hyperparameter settings for the CodeTracer watermarking framework. The parameters are organized by their functional categories: watermark configuration controlling detectability, model architecture defining the watermark policy network structure, training configuration governing the optimization process, and reward configuration balancing the various objectives.

| Parameter | Value | Description |
|---|---|---|
| *Watermark Configuration* | | |
| Watermark bias ($\delta$) | 2.0 | Logit adjustment magnitude for green tokens |
| Green list ratio ($\gamma$) | 0.5 | Proportion of vocabulary in green list |
| Z-score threshold | 4.0 | Statistical threshold for watermark detection |
| *Model Architecture* | | |
| Base model | OpenCoder-1.5B-Instruct | Foundation model for code generation |
| Model dimension ($d_{\text{model}}$) | 512 | Hidden dimension for watermark policy network |
| Transformer layers | 6 | Number of transformer encoder layers |
| Attention heads | 8 | Multi-head attention heads per layer |
| Feed-forward dimension | 2048 | Intermediate dimension in feed-forward networks |
| *Training Configuration* | | |
| Learning rate | 1.0e-5 | Initial learning rate for optimization |
| LR scheduler | Cosine with min rate | Decay schedule with minimum rate of 0.1 |
| Batch size | 8 | Sequences per device for training |
| Gradient accumulation | 4 | Steps between gradient updates |
| Training steps | 500 | Total optimization steps |
| Warmup ratio | 0.03 | Proportion of steps for learning rate warmup |
| KL coefficient ($\beta_{\text{KL}}$) | 0.0 | Weight for KL divergence regularization |
| Number of generations | 8 | Code completions per prompt during training |
| Temperature | 1.0 | Sampling temperature for generation |

where $\alpha > 1$ (typically 2-5) creates a stronger penalty for red token selection when watermarking is active. This asymmetric design incentivizes the model to be more selective about when to apply watermarking ($w = 1$), only doing so when it has high confidence in successfully biasing toward green tokens. For less certain positions, the model learns to disable watermarking ($w = 0$), which receives a neutral reward. This approach significantly reduces false-positive detection rates while maintaining high true-positive rates.

### B.5. Limitations of Using LLMs as Watermark Policy Networks

While using pre-trained LLMs directly as watermark models might seem appealing, we identified several critical limitations that justify our specialized architecture. First, **distribution alignment** presents a fundamental challenge: LLM token distributions naturally align with the target model distribution, causing "green" token lists to predominantly contain frequently used tokens. This similarity makes distinguishing between unwatermarked and watermarked sequences difficult, as both produce similarly high z-scores during detection. Second, **computational efficiency** concerns arise as large-scale LLMs with billions of parameters introduce significant processing latency compared to our specialized watermark policy, which operates with orders of magnitude fewer parameters while maintaining task-appropriate performance.

Third, **context length mismatch** creates architectural incompatibilities, as LLMs are optimized for extended contexts (thousands of tokens), while our watermark policy intentionally operates with minimal contextual information (typically just a few tokens) to enable efficient decision-making at each generation step. Fourth, **entropy prediction calibration** poses practical challenges because LLMs produce poorly calibrated uncertainty estimates for short contexts, particularly at document beginnings, which compromises the reliability of watermark decisions in these critical positions. Our purpose-built watermark policy network addresses these limitations with a lightweight, specialized architecture that achieves superior watermarking performance while maintaining the natural fluency of generated code.

## C. Implementation Details

Table 6 presents the comprehensive hyperparameter configuration for our CodeTracer framework, categorized by functional components to facilitate reproducibility and highlight critical design decisions across our experimental setup.

*Table 7.* Prompt-free comparison with Gumbel-max watermarking on HumanEval.

| Method | Pass@1 | Pass@10 | AUROC | TPR@5%FPR |
|---|---|---|---|---|
| Gumbel-max | 25.85 | 29.48 | 77.84 | 25.00 |
| CodeTracer-1 | 62.65 | 77.11 | 77.71 | 32.32 |

*Table 8.* Full-information SWEET reference comparison on HumanEval. This privileged setting is diagnostic and is not the main prompt-free and base-model-free detection setting.

| Method | Pass@1 | Pass@10 | AUROC | TPR@5%FPR |
|---|---|---|---|---|
| SWEET full-info | 61.77 | 75.45 | 83.10 | 42.68 |
| CodeTracer* full-info | 64.13 | 76.39 | 93.38 | 61.87 |

## C.1. Datasets

We evaluate on standard code generation benchmarks that span multiple programming languages and tasks. The primary evaluation uses HumanEval (Chen et al., 2021) and MBPP (Austin et al., 2021), which provide comprehensive Python programming challenges with associated test cases and reference implementations.

## C.2. Baselines

We compare against two categories of methods: post-hoc detection and active watermarking. Post-hoc detection methods preserve original generation and include zero-shot approaches: logp(x), LogRank (Gehrmann et al., 2019), DetectGPT (Mitchell et al., 2023), and GPTZero (Tian & Cui, 2023). Active watermarking methods include WLLM (Kirchenbauer et al., 2023a) and EXP-edit (Kuditipudi et al., 2023), which operate under the practical constraints. We also include SWEET (Lee et al., 2023), though comparisons are not fair since it requires access to the original LLM and prompts for entropy calculation. We adapt SWEET to use a practical setting for fair comparison with other baselines. We thoroughly tune the hyperparameters for each baseline following their setup.

**DetectGPT.** For the DetectGPT implementation, we used T5-3B as our model. Following the original DetectGPT paper and SWEET (Mitchell et al., 2023; Lee et al., 2023), we set the span length to 2 words and applied masking to 20% of the text. For each test, we generated 100 perturbations to ensure robust detection.

**SWEET.** For the SWEET baseline implementation, we conducted a systematic hyperparameter search for the entropy threshold, exploring values from 0.3 to 1.2 with increments of 0.3. Our experiments revealed optimal performance with entropy thresholds of 1.2 and 0.9 for the HumanEval dataset and 0.3 and 0.6 for MBPP. During detection, SWEET needs to use the original LLM, prompt, and complete generation sequence to compute entropy. This requirement severely restricts its practical applicability in real-world scenarios. For practical deployment, an ideal solution would depend solely on the code snippet itself, as access to the generation context is typically unavailable. Thus, we adapt SWEET to use fixed context windows (vs. full generation history and original LLM) for practical and fair comparison with other baselines. We select two entropy thresholds for SWEET to demonstrate the trade-off between watermark detectability and code functionality.

**EXP-edit.** We evaluate EXP-edit (Kuditipudi et al., 2023) following the methodology of Lee et al. (2023). In our experiments, we set temperature=0.2 and top-p=0.95. We systematically explore block sizes (20 tokens), key sequence lengths (100), resample sizes (50 runs), and edit distance thresholds ($\gamma = 0.0$). Through extensive parameter tuning, we determine the optimal configuration: key length 100, block size 20, 50 sampling runs, and detection threshold 0.1, which balances watermark detection reliability with code functionality.

**Additional Baseline Comparisons.** Table 7 compares CodeTracer-1 with Gumbel-max watermarking under the shared prompt-free setting. Gumbel-max reaches comparable AUROC, but its functional correctness is much worse under this setting. Table 8 reports a full-information reference comparison with SWEET. This is a privileged diagnostic setting requiring access to information outside our main prompt-free and base-model-free threat model.

*Table 9.* Ablation of the watermark policy context window on HumanEval.

| $c$ | Pass@1 | Pass@10 | AUROC | TPR@5%FPR |
|---|---|---|---|---|
| 1 | 66.40 | 78.09 | 62.28 | 14.02 |
| 2 | 62.65 | 77.11 | 77.71 | 32.32 |
| 4 | 62.65 | 75.80 | 70.96 | 21.95 |
| 8 | 65.58 | 79.61 | 69.44 | 11.59 |

### C.3. Model Architecture Specifications

Our model implements a transformer-based architecture optimized for code analysis and generation. The model uses pre-norm design with layer normalization before the attention and feed-forward computations to enhance training stability. It consists of four main components:

**Embedding Layer**: Combines token embeddings ($W_{\text{te}} \in \mathbb{R}^{|\mathcal{V}| \times d_{\text{model}}}$) and position embeddings ($W_{\text{pe}} \in \mathbb{R}^{c \times d_{\text{model}}}$), where $d_{\text{model}} = 512$ and $c = 2$ (context window size), processed through a dropout layer with rate 0.2.

**Transformer Encoder**: 6 layers with 8 attention heads per layer.

**Feed-Forward Networks**: Dimension expansion to 2048 before projection back to $d_{\text{model}}$.

**Output Layer**: Final layer normalization followed by linear projection to $|\mathcal{V}| + 1$ dimensions.

**Hyperparameter Configuration.** We conducted extensive hyperparameter tuning to optimize our watermarking system's performance. Table 6 presents our final hyperparameter configuration. For model-specific parameters, we use the OpenCoder-1.5B-Instruct base model with BF16 precision and flash attention 2 implementation for optimal performance. The watermark policy network uses a context window of 2 tokens to determine watermarking decisions, which balances detection effectiveness and computational efficiency. The watermark strength parameters include a green list ratio $\gamma = 0.5$, establishing an equal distribution between green and red token sets. The watermark binary decision threshold is set at 0.5, providing a balanced approach for selective watermarking. For detection, we employ a z-score threshold of 4.0, which ensures high precision in distinguishing watermarked code from non-watermarked code while maintaining a low false-positive rate.

**Context Window Ablation.** Table 9 reports the effect of varying the watermark policy context window on HumanEval. The base LLM always receives the full prompt and generated prefix; only the lightweight watermark policy context changes. We use $c = 2$ in all main experiments because it gives the best utility-detectability trade-off. Larger contexts maintain similar functionality but reduce detectability, likely because the prompt-free detector must reconstruct watermark decisions from generated code alone and longer contexts introduce more context-specific, less repeatable watermarking decisions.

### C.4. Training Details

We implement reinforcement learning training using GRPO with several customizations for the code watermarking task. Our training pipeline processes a mix of prompts from the `open-r1/verifiable-coding-problems-python` dataset, `HumanEval`, and `MBPP`. The motivation is to make the model familiar with the distribution of these dataset instructions, which are representative code examples for code generation tasks.

The watermark policy model optimization uses a learning rate of $1.0 \times 10^{-5}$ with a cosine scheduler including a minimum learning rate of 0.1. We implement a short warmup period (3% of total steps) to stabilize early training. For regularization, we set $\beta = 0.0$ for the KL divergence term, allowing the watermark policy to explore freely while process-based rewards provide necessary constraints. The training process runs for 500 steps with a batch size of 8 per device and gradient accumulation every 4 steps. For each prompt, we generate 8 different code completions to provide robust training signals. The maximum prompt length is limited to 256 tokens, while completions can extend up to 1024 tokens to accommodate various code generation tasks. During inference, we use a temperature of 1.0 to maintain natural generation characteristics.

**Computation Resources** All experiments were conducted on one NVIDIA A100 GPU with 80GB memory, using BF16 precision to maximize computational efficiency. The watermark policy network training requires significantly fewer resources than full LLM training. For inference and evaluation, a single A100 GPU is sufficient to process the benchmark

test cases. The lightweight nature of our watermark policy ensures minimal computational overhead during both training and deployment compared to LLM-based watermarking approaches. Our implementation uses PyTorch with flash attention 2 for optimized transformer operations. To ensure reproducibility, we set a random seed of 42 across all experiments. The relatively modest computational requirements make our approach practical for real-world deployment scenarios, as the watermark policy can be integrated with various LLMs without substantial infrastructure demands.

## D. Security Discussion

The security of CodeTracer's watermarking framework derives from the complexity of our learning-based approach and the semantic awareness of our watermark model. This section analyzes the security properties of our approach compared to traditional watermarking methods.

Our transformer-based watermark model provides strong security through its complex neural representations. Unlike static watermarking schemes that apply fixed rules, our architecture learns context-dependent watermarking decisions that adapt to the syntactic and semantic structure of code. The high-dimensional parameter space of our 6-layer transformer model creates a complex mapping between code contexts and watermarking decisions that resists reverse engineering. This neural complexity means that even with access to watermarked outputs, attackers face significant computational barriers to inferring the underlying decision boundaries that govern watermark placement and token partitioning.

The selective application of watermarking through the binary decision mechanism (w) adds another layer of security. Our model strategically applies watermarking only in positions where it can effectively bias toward green tokens without compromising code functionality. This selective approach creates an irregular watermarking pattern that varies with code structure and semantics, making statistical detection more challenging for attackers attempting to identify and remove watermarks.

In contrast, traditional watermarking approaches relying solely on hash functions or fixed vocabulary partitioning remain vulnerable to statistical attacks. Attackers can analyze token distribution patterns across multiple watermarked samples to eventually infer the underlying partitioning scheme. Our framework avoids this vulnerability by creating position-specific, semantically-informed token partitions that vary based on code context.

For a successful attack, adversaries would need to simultaneously: (1) reverse-engineer the neural network's complex decision boundaries, (2) understand the dynamic token partitioning strategy across different code structures, and (3) develop a method to generate valid code that avoids watermarked patterns without compromising functionality. This multi-faceted security approach provides robust protection against both detection evasion and watermark forgery in practical deployment scenarios.

