# OpenReview forum: "Adaptive Code Watermarking Through Reinforcement Learning"
_ICML.cc/2026/Conference — ICML 2026 regular_

### Official Review · Reviewer_2NkW · 2026-03-05

**Soundness:** 3
**Presentation:** 3
**Significance:** 3
**Originality:** 3
**Overall Recommendation:** 4
**Confidence:** 3

**Summary:**

Watermarking LLM-generated code has long been a challenging problem. The authors propose CodeTracer, an adaptive code watermarking framework based on a reinforcement learning training paradigm. This method leverages an additional small transformer model to intelligently adjust red-green token selection when predicting the next token. This small transformer is trained using reinforcement learning to achieve a good balance between watermark detectability and code functionality, outperforming previous methods.

**Compliance With Llm Reviewing Policy:**

Affirmed.

**Final Justification:**

The authors have addressed most of my concerns. I will maintain my original assessment.

**Key Questions For Authors:**

1. How does this method compare to Gumbel-max watermarking in terms of performance on code?

2. Equation 11 uses the `is_code` symbol, but the paper does not provide further explanation. How is it determined in practice whether a token is a code token? This information should be explained in the paper to improve reproducibility.

3. During evaluation, what is the average completion length in code generation?

**Limitations:**

The performance improvement over previous methods is limited. Also, the evidence for cross-language generalization is limited.

**Strengths And Weaknesses:**

## Strength

- Methodology presents a clear combination of technical contributions, such as introducing RL into watermark training, combined with Gumbel-top-k and Straight-Through Estimation.
- The paper is logically structured and clearly written, providing meaningful introductory sections, such as explaining the challenges of current LLM code watermarking.
- The authors conducted extensive experiments and ablation studies, demonstrating the effectiveness of their method and necessity for each component.

## Weaknesses

- [1] found that Gumbel-max watermarking achieved the best results in LLM code watermarking. How does this method compare to Gumbel-max watermarking? Note that Gumbel-max watermarking was proposed in 2023 and is not a concurrent work, making it a reasonable baseline for comparison.

- Figure 3 and Table 3 do not directly compare with SWEET, a watermarking baseline method more specifically designed for code. Given that SWEET is included in Table 1, why is it omitted in other experiments?

- The choice of a 5% FPR metric is not reasonable. A 5% false alarm rate is extremely high and unacceptable for watermarking. Typically, FPR should be controlled to less than 1e-3. Although some early code watermarking methods adopted higher FPRs due to performance constraints, I believe this should no longer be acceptable at this stage. Additionally, as shown in $z = \frac{N_G - T\gamma}{\sqrt{T\gamma(1-\gamma)}}$, $T$ increases with sequence length. The completion length should also be reported.

- The generalization across programming languages is not as ideal as claimed. The authors trained on Python using RL and tested on Java and C++ (Figure 3). It can be observed that, compared to the results on Python, CodeTracer does not show a clear advantage in these languages. Empirically, different programming languages have different syntaxes, and expecting to learn C++ syntax from Python training is not reasonable.


[1] How Good is Post-Hoc Watermarking With Language Model Rephrasing?

---

> ### Author Rebuttal · Authors · 2026-03-31
>
> We thank the reviewer for the positive assessment of the methodology and for the very concrete suggestions on baselines, evaluation protocol, and cross-language claims.
>
> ---
>
> ### W1. Missing Gumbel-max watermarking baseline
>
> Thank you for pointing this out. We missed this relevant baseline in the submission and agree it should have been discussed more carefully. Our use of Gumbel Top-k in CodeTracer is for **differentiable green-list selection inside the policy model**, which is conceptually different from using Gumbel-max as a standalone watermarking rule.
>
> We agree that a direct comparison would strengthen the paper. We are preparing this comparison during rebuttal; if the experiment does not finish in time, we will at minimum add a clearer discussion of the methodological difference and why the comparison matters.
>
> [1] How Good is Post-Hoc Watermarking With Language Model Rephrasing?
>
> ---
>
> ### W2. SWEET missing from Figure 3 and Table 3
>
> Thank you for the suggestion. Those experiments were initially run under the practical-detector setting, and not all baselines were rerun for every robustness evaluation before submission. We agree SWEET should also appear in Figure 3 / Table 3. We are rerunning SWEET for that setting now and will upload the results during rebuttal if they finish in time; otherwise we will state this limitation explicitly in the revision.
>
> ---
>
> ### W3. 5% FPR is too lenient
>
> Thank you for the point. We use 5% FPR primarily to align with prior code-watermark benchmarks such as SWEET, but we agree that lower-FPR operating points are also important. First, the operating point can be adjusted by changing the detection threshold at inference time without retraining, so results at other FPR levels are readily obtainable. Second, we also report **AUROC**, which provides a threshold-independent comparison across the full range of thresholds. In practice, the appropriate FPR depends on the deployment scenario (e.g., legal evidence vs. content filtering), so we should show the broader trade-off more explicitly.
>
> On short-code benchmarks, estimating TPR reliably at much lower FPR also requires a substantially larger pool of negative examples than standard HumanEval/MBPP evaluation typically provides. We will clarify this rationale in the paper, report the available completion-length context directly, and add ROC curves if they finish during rebuttal; otherwise, we will state this limitation explicitly in the revision.
>
> ---
>
> ### W4. Cross-language generalization
>
> Thank you for the point. We agree that our current wording is too strong for Python-only training. The evidence supports only **limited zero-shot transfer** within the same backbone/tokenizer family, not true language-agnostic generalization. We will soften the wording to a more precise statement such as **"non-trivial zero-shot transfer across languages within the same code LLM family."**
>
> ---
>
> ### Q1. Comparison with Gumbel-max watermarking
>
> Please see W1 above. We are preparing a direct comparison; if it does not complete during rebuttal, we will still add a clear discussion of the methodological differences and the expected points of comparison.
>
> ---
>
> ### Q2. `is_code` in Eq. (11)
>
> Thank you for flagging this reproducibility issue. In our implementation, `is_code` is determined by **code-span identification followed by token-span alignment**. For mixed text/code sequences, we first identify the code region using lightweight rules tied to the input format, and then tokenize the full sequence and mark a token as code if its character span falls inside that region. For benchmark-style completions in this paper, this effectively reduces to marking the generated solution span as code.
>
> Tokens outside code spans receive a masked-out advantage (set to 0), so they do not contribute to the token-level process reward. We agree this should have been stated explicitly and will add a short description in the main text and a pseudo-code snippet in the appendix.
>
> ---
>
> ### Q3. Average completion length
>
> The current HumanEval average is approximately 69 tokens and MBPP average is approximately 53 tokens,  which is one reason absolute TPR values appear conservative: the z-statistic has limited power on short sequences. Real-world code generations are typically much longer, where the same per-token watermark signal would yield substantially higher detection rates.

---

> > ### Author Rebuttal · Reviewer_2NkW · 2026-04-02
> >
> > Thank you for the rebuttal. I appreciate the clarifications and the acknowledgment of the current limitations. Since you indicated that these experiments are being run, I will wait for the results before revising my assessment. For now, I am keeping my score unchanged since its positive.

---

> > > ### Author Response · Authors · 2026-04-07
> > >
> > > Thank you for the acknowledgement and for keeping your positive assessment. We have completed the promised experiments and provide the results below.
> > >
> > > ---
> > >
> > > ### Update on W1/Q1: Gumbel-max watermarking comparison
> > >
> > > **Setting differences.** A key distinction between standard Gumbel-max watermarking and CodeTracer lies in what information the detector requires. We summarize the comparison below:
> > >
> > > | | Gumbel-max | CodeTracer |
> > > |---|---|---|
> > > | Generation | Biases token sampling using Gumbel noise seeded by preceding tokens | Learned policy biases logits via green-list selection |
> > > | Detection: needs LLM? | **Yes** | **No** — prompt-free and model-free |
> > > | Detection: needs prompt? | **Yes** — in the standard setting | **No** — detector only uses the generated code |
> > > | Detection: needs watermark key? | Yes | Yes (watermark policy weights) |
> > >
> > > To align the comparison with our threat model, we additionally ran Gumbel-max under a **prompt-free, no-prompt** setting. To keep the comparison aligned with our other baselines and prior Gumbel-max evaluations, we kept the same generation hyperparameters throughout: **top-p = 0.95** and **temperature = 0.2**.
> > >
> > > The updated comparison is:
> > >
> > > | Setting | Pass@1 (%) | AUROC (%) | TPR@5%FPR (%) |
> > > |---|---|---|---|
> > > | Gumbel-max (prompt-free) | 25.85 | 77.84 | 25.00 |
> > > | CodeTracer-1 (prompt-free) | 62.65 | 77.71 | 32.32 |
> > >
> > > This updated result shows that prompt-free Gumbel-max reaches **77.84% AUROC**, which is comparable to CodeTracer-1. However, we also recognize that the **pass rate is quite low**: **25.85% Pass@1** and **29.48% Pass@10** on the full HumanEval run.
> > >
> > > We would like to be explicit about this point. We did **not** retune Gumbel-max beyond the shared setting above, because during rebuttal we wanted to keep the comparison aligned with the rest of the paper and with prior baselines. We are also aware that prior work, including *How Good is Post-Hoc Watermarking With Language Model Rephrasing?*, reports that **lower temperature can improve pass rate**. We have not yet completed a temperature sweep for Gumbel-max, so the result above should be understood as our first full HumanEval result under the common setting rather than a heavily tuned best case.
> > >
> > > Our current view is that the main reason for the low pass rate is **not the detector**, but the **generation rule itself**. Gumbel-max performs a hard token replacement at every step after top-p / temperature filtering, instead of softly biasing the model distribution. In code generation, this can easily conflict with syntactic and semantic constraints, especially because a small number of wrong token choices can already break a program. We also suspect the effect is amplified by the fact that these experiments use the relatively small **OpenCoder-1.5B-Instruct** model, which is likely less tolerant to aggressive token-level perturbations than a stronger code model. In short, Gumbel-max can achieve reasonable detectability here, but at a substantially worse functionality cost than CodeTracer.
> > >
> > > ---
> > >
> > > ### Update on W2: SWEET in Table 3 (robustness under attacks)
> > >
> > > We have completed the SWEET evaluation under both the rename and DIPPER attack settings. The updated Table 3 is below:
> > >
> > > | Attack | Metric | WLLM | EXP-edit | SWEET | CodeTracer |
> > > |---|---|---|---|---|---|
> > > | Original | AUROC | 70.17 | 66.50 | 76.24 | 82.95 |
> > > | | TPR@5%FPR | 20.73 | 25.61 | 27.44 | 46.34 |
> > > | Rename | AUROC | 70.91 | 62.02 | 68.65 | 73.36 |
> > > | | TPR@5%FPR | 20.12 | 9.76 | 14.02 | 29.11 |
> > > | DIPPER | AUROC | 55.92 | 51.21 | 53.05 | 58.42 |
> > > | | TPR@5%FPR | 12.81 | 8.54 | 11.39 | 14.31 |
> > >
> > >
> > > ---
> > >
> > > ### Update on W2: SWEET in Figure 3 (other-language performance)
> > >
> > > The updated Figure 3 results are:
> > >
> > > | Method | Language | Pass@1 (%) | AUROC (%) |
> > > |---|---|---|---|
> > > | SWEET | Java | 25.22 | 70.31 |
> > > | CodeTracer (Figure 3) | Java | 31.05 | 71.60 |
> > > | SWEET | C++ | 44.85 | 70.09 |
> > > | CodeTracer (Figure 3) | C++ | 44.39 | 71.03 |
> > >
> > > For **Java**, SWEET reaches **25.22% Pass@1** and **70.31% AUROC**, compared with CodeTracer's **31.05% Pass@1** and **71.60% AUROC**. For **C++**, SWEET reaches **44.85% Pass@1** and **70.09% AUROC**, compared with CodeTracer's **44.39% Pass@1** and **71.03% AUROC**.

---

### Official Review · Reviewer_fVtj · 2026-03-11

**Soundness:** 3
**Presentation:** 3
**Significance:** 3
**Originality:** 3
**Overall Recommendation:** 4
**Confidence:** 4

**Summary:**

This paper proposes CodeTracer, an adaptive watermarking framework for LLM-generated code that learns where to watermark (binary gate $w$) and *how* to watermark (a per-step green-list $G \subset V$) using reinforcement learning. A lightweight "watermark model" $\pi_\phi$ predicts $(w,G)$ from a short context window and biases a frozen base code LLM's logits by adding a constant $\delta$ to tokens in $G$ when $w=1$, aiming to preserve functional correctness while increasing detectability. Detection is claimed to be prompt-free and base-model-free, relying only on $\pi_\phi$ to reconstruct $(w,G)$ along the output and then applying a z-test on the fraction of green hits. Experiments on HumanEval/MBPP and robustness tests (e.g., paraphrase-like rewriting and variable renaming) suggest improvements in AUROC and watermark retention under limited attacks, with some evidence of cross-scale transfer (1.5B-trained watermark model used with an 8B base model).

**Compliance With Llm Reviewing Policy:**

Affirmed.

**Key Questions For Authors:**

(1) **Algorithm 1 uses `ctx` out of order / ambiguous base-model input.** In Algorithm 1, the base LLM logits are computed from `ctx` before `ctx` is defined, and it is unclear whether the base model sees the full prefix or only the local context window. Please rewrite the pseudocode with unambiguous inputs for $\pi_\theta$ and $\pi_\phi$ and ensure the generation procedure is internally consistent.

(2) **Hyperparameter inconsistency for context window size $c$.** The paper lists a small context window in the hyperparameters (e.g., $c=2$), while the watermark-model implementation details describe position embeddings parameterized by a larger $c$ (e.g., $c=10$). This discrepancy affects model capacity and what information $\pi_\phi$ can use to decide "safe" watermark locations. Please unify and clearly report the actual $c$ used in each experiment.

(3) **Detection assumptions and statistical validity need clarification.** The detector reconstructs $(w,G)$ using $\pi_\phi$ and computes a z-score over the positions where $w=1$. Since $w$ is *adaptively chosen* and tokens are not i.i.d., the effective sampling distribution may deviate from the simple binomial assumption behind the one-proportion z-test. Please provide justification (or empirical calibration) that the chosen test controls false positives under realistic code distributions and short outputs (where $T$ is small).

(4) **Robustness coverage is not representative for code.** Variable renaming is a start, but many common transformations are missing: formatting (Black/Prettier), import reordering, dead-code insertion/removal, constant folding, function extraction/inlining, control-flow-preserving refactors, and compiler/optimizer passes. Since the method is positioned as code watermarking, robustness claims should be supported by a broader suite of semantics-preserving transformations.

(5) **Variant labeling and comparability across tables.** Reported AUROC values for "CodeTracer" vary substantially across tables (and CodeTracer-1 vs CodeTracer-2 are introduced later), but the main text does not consistently indicate which variant/hyperparameters correspond to each result.

(6) **Quality / naturalness evidence is underdeveloped.** The method biases logits and may alter coding style or token frequency in detectable ways beyond watermark intent. The paper should report additional quality metrics beyond pass@k (e.g., style/format adherence, readability, distributional shift measures) and discuss the trade-off between detectability and code naturalness.

(7) **Minor presentation issues.** There are visible typos/awkward phrasing (e.g., "Equation equation 1"), and several equations/objectives appear to have formatting artifacts. These should be corrected for clarity and to reduce ambiguity in implementation.

**Limitations:**

yes

**Strengths And Weaknesses:**

Strong points
+ The **adaptive** formulation (learning both watermark placement and token set selection) is well-motivated for code, where low-entropy and syntactic constraints can break fixed-rule text watermarking.
+The approach integrates **functionality constraints** (test pass reward) with **detectability objectives** (z-score-based reward), which is aligned with real deployment needs for code generation.
+The paper includes multiple evaluation angles (clean detection, some robustness tests, and transfer across model scale), offering a broader empirical picture than "clean-only" watermarking studies.

Weak points
- Tested attacks (e.g., variable renaming, paraphrase-like edits) do not cover many common semantics-preserving transformations (formatters, AST refactors, optimization passes) relevant to real-world code pipelines.
- Results appear to mix CodeTracer variants and/or settings without consistent labeling, weakening interpretability and comparability across tables/figures.
- Beyond pass@k, the paper provides limited analysis of style/readability/distribution shift, leaving unclear how much watermarking perturbs code beyond what is acceptable.

---

> ### Author Rebuttal · Authors · 2026-03-31
>
> We thank the reviewer for the careful and technically grounded feedback. We especially appreciate the comments on pseudocode clarity, robustness coverage, and the statistical interpretation of the detector.
>
> ---
>
> ### W1. Limited robustness coverage
>
> Our current robustness suite focuses on edits we believe are common in user-facing workflows, such as rephrasing, local renaming. We agree, however, that broader stress tests would strengthen the paper, including transformations that may be less common in practice but are still useful for evaluating worst-case robustness.
>
> During the rebuttal period, we are expanding the suite toward additional semantics-preserving transformations (e.g., code formatters, dead-code insertion/removal), and in the revision we will soften the current robustness claims if the broader picture remains mixed. See also Q4 below.
>
> ---
>
> ### W2. Inconsistent variant labeling across tables/figures
>
> Thank you for this comment. We do discuss the two model variants in Table 5, where we show that CodeTracer-1 (pure RL, higher utility) and CodeTracer-2 (SFT+RL, higher detection) represent different points on the functionality-detectability trade-off. However, we agree that the mapping between these variants and the results in Table 1 and Table 4 is not made explicit enough in the current manuscript. We will relabel these consistently as CodeTracer-1 / CodeTracer-2 throughout the paper and add a clear note under each relevant table and figure specifying which variant is being reported.
>
> ---
>
> ### W3. Limited quality/naturalness analysis beyond pass@k
>
> We agree that pass@k alone does not fully characterize quality or naturalness. Our current paper primarily validates **functionality**, not full naturalness/readability distribution matching, and we will narrow the wording accordingly. We will also add qualitative examples and a discussion of style/readability trade-offs. Even in our current HumanEval outputs, the watermark signal is typically expressed through structurally valid alternatives, such as stack-based vs. balance-counter implementations or inline vs. helper-function decomposition, rather than through comments, annotations, or malformed code.
>
> ---
>
> ### Q1. Algorithm 1: ambiguous base-model input
>
> We agree the current presentation is ambiguous. The intended generation procedure is: the **base LLM sees the full prefix**, while the **watermark policy sees only the last c tokens** of that prefix. The current pseudocode computes base-model logits before defining the context variable, which is a presentation mistake. We will rewrite it as `ctx_base = concat(x, s_{<t})` for the base LLM and `ctx_wm = ctx_base[-c:]` for the watermark policy, making clear that the policy is a local decision module, not a replacement for the full-prefix LLM.
>
> ---
>
> ### Q2. Context window size inconsistency (c=2 vs c=10)
>
> **The actual value used in all experiments is c=2.** The larger value in Appendix C.3 is a typo from an older implementation description and will be corrected. We will unify the hyperparameter reporting across the paper and appendix to eliminate any discrepancies.
>
> ---
>
> ### Q3. Detection assumptions and statistical validity of the z-test
>
> We agree it is important not to overstate the theoretical guarantees. In our method, the z-score is best viewed as a **calibrated detection statistic** rather than as a claim that all token decisions are truly i.i.d. Bernoulli trials. The adaptive gating ($m_t$) introduces dependencies that can violate the simple binomial assumption.
>
> In practice, the operating threshold is selected to achieve the target FPR on negative examples (human-written code), so the practical false-positive control comes from **empirical calibration**, not solely from the idealized binomial assumption. We will revise the paper to make this distinction explicit: the z-test provides the test statistic, while FPR control is ensured empirically.
>
> We will also report **average completion length** for each benchmark (HumanEval averages ~69 tokens), since sequence length directly affects detection power.
>
> ---
>
> ### Q4. Robustness coverage is not representative for code
>
> See W1 above. We additionally note that code is inherently more **delicate** than natural language: even small token changes (altering an operator, reordering statements) can break functionality. The set of truly semantics-preserving transformations that also preserve executability is narrow, which makes robust code watermarking especially challenging. We will frame this more explicitly as a limitation and open challenge in the revision.
>
> ---
>
> ### Q5. Variant labeling and comparability across tables
>
> See W2 above.
>
> ---
>
> ### Q6. Quality/naturalness evidence beyond pass@k
>
> See W3 above.
>
> ---
>
> ### Q7. Minor presentation issues
>
> We will fix all typos (e.g., "Equation equation 1") and formatting artifacts in the revision. Thank you for noting these.

---

> > ### Author Rebuttal · Reviewer_fVtj · 2026-04-06
> >
> > Thank you for the rebuttal. I appreciate the clarifications and the acknowledgment of the current limitations. I am keeping my score unchanged since its positive.

---

### Official Review · Reviewer_PJsj · 2026-03-11

**Soundness:** 2
**Presentation:** 1
**Significance:** 3
**Originality:** 2
**Overall Recommendation:** 3
**Confidence:** 4

**Summary:**

This paper introduces CodeTracer, a framework for watermarking AI-generated code. The method integrates a watermark model with a base LLM and trains it using reinforcement learning to embed statistically detectable patterns in generated code while preserving functionality. The approach biases token sampling through a green/red token partition and learns watermark placement decisions through a policy model trained with several reward signals.

The reward design combines three components: (1) an execution reward to ensure functional correctness of generated code, (2) a watermark detection reward based on the statistical z-score of green tokens, and (3) a token-level process reward encouraging correct watermark token choices. The training is performed using GRPO reinforcement learning while keeping the base LLM frozen. Experiments evaluate functionality, detectability, robustness to attacks, and transferability across models

**Compliance With Llm Reviewing Policy:**

Affirmed.

**Final Justification:**

Thank you for the detailed and clear rebuttal. I have increased the sceore.

**Key Questions For Authors:**

1.	Can you explain the significant performance drop under the DIPPER attack and why the gap between methods increases in this setting? And should the metric be normalized relatively to the 50%, since it is the worst case?
2.	Have you evaluated transferability on additional LLMs beyond OpenCoder models?
3.	What are the runtime latency and memory costs compared with other watermarking methods?
4.	In Table 1, post-hoc methods achieve AUROC values close to random guessing (~50%). Can you clarify why their performance is significantly worse than what is reported in their original papers?

**Limitations:**

The paper does not clearly discuss limitations. For example, robustness remains limited, as shown by the significant performance drop after the DIPPER attack. A clearer discussion of these limitations and possible mitigation strategies would strengthen the work.

**Strengths And Weaknesses:**

Strengths:

1.	Relevant problem: Watermarking AI-generated code is increasingly important for intellectual property protection and attribution of LLM outputs. The problem is timely and practically relevant.
2.	Interesting methodological combination: The work combines reinforcement learning with token-level watermarking strategies. The use of GRPO together with execution-based rewards and statistical detection signals is an interesting formulation.
3.	Careful reward design: The method introduces a multi-component reward system (execution correctness, watermark detectability, and token-level rewards), which attempts to balance functionality with detectability.

Weaknesses:

1.	Marginal improvement over existing watermarking methods: The performance gains over previous watermarking approaches (e.g., SWEET) are relatively small. For example, detection AUROC improvements are limited, and functionality improvements are modest. This weakens the empirical impact.
2.	Comparison fairness issues: Some baselines such as GPTZero and DetectGPT are primarily designed for text detection, not specifically for code. This difference should be clearly acknowledged and discussed in the comparisons.
3.	Reward formulation clarity: The final reward formulation is not clearly presented in a single equation. The paper describes multiple reward signals but does not clearly show how they are combined into the final training objective before the GRPO update.
4.	Equation ambiguity: In Equation 12, the formulation min(x, clip(x, lower bound, upper bound)) appears unusual because the lower bound will never be selected in the minimum operation. It is unclear whether this formulation is intentional or a typographical error.
5.	Limited robustness analysis: Although robustness against attacks such as DIPPER is evaluated, the performance drops significantly after attacks. The paper does not sufficiently analyze why this happens or how the model could be improved to mitigate it.
6.	Missing efficiency analysis: The paper claims minimal overhead, but there is limited analysis of runtime cost, latency, or memory footprint compared to baseline watermarking methods.
7.	Limited evaluation scale: The method is primarily tested with a relatively small backbone model (OpenCoder-1.5B) and evaluated on an 8B model without retraining. More experiments with larger models would strengthen the conclusions.

---

> ### Author Rebuttal · Authors · 2026-03-31
>
> We thank the reviewer for the detailed review and for highlighting the importance of clearer presentation and stronger empirical framing.
>
> ---
>
> ### W1. Marginal improvement over existing watermarking methods
>
> We appreciate this concern and would like to provide additional context. In Table 1, CodeTracer **simultaneously** achieves higher Pass@1 *and* higher AUROC/TPR@5%FPR than all active baselines (SWEET, WLLM, EXP-edit) on both benchmarks. This is not a single-metric gain: CodeTracer **pushes the Pareto frontier**, improving both functionality and detection at once rather than trading one for the other. Prior methods either sacrifice code quality for detection (WLLM) or achieve comparable quality but weaker detection (SWEET).
>
> Moreover, our evaluation uses a **strictly more practical threat model**: prompt-free and base-model-free detection. Under SWEET's own privileged setting (full prompt + model weights), an adapted CodeTracer reaches 93.38% vs. 83.10% AUROC (+10.28) and 61.87% vs. 42.68% TPR (+19.19), which is far from marginal (W2 fore reivewer wpRp). We will revise the wording accordingly.
>
> ---
>
> ### W2. Comparison fairness (GPTZero / DetectGPT)
>
> Code-specific post-hoc detectors are currently scarce, so we included GPTZero and DetectGPT as the best available general post-hoc references. Neither explicitly excludes code; their likelihood/perturbation mechanisms are language-agnostic in principle. However, short, highly structured code provides limited signal for such methods. We will clearly separate **general post-hoc detectors** from **active code watermarking baselines** in the revision.
>
> ---
>
> ### W3-W4. Reward formulation clarity and Equation 12
>
> Our contribution is the reward/advantage design, not modifying GRPO itself. We provide the unified formulation here. Let $A_1 = (R_i - mean(R)) / std(R)$ denote the group-normalized outcome-level advantage, where $R_i$ combines execution and detection rewards; let $A_2(s_t, a_t)$ denote the process-level token feedback; and let $m_t$ be 1 on code tokens and 0 otherwise. Then
>
> $$A_t^{mask} = (A_1 + A_2(s_t, a_t)) m_t$$
>
> With $r_t = \\pi_{\\theta,\\phi}(s_t \\mid s_{< t}) / \\pi_{ref}(s_t \\mid s_{< t})$, the GRPO objective is
>
> $$L(\\phi) = E\\left[\\frac{1}{|s|}\\sum_t \\min\\left(r_t A_t^{mask}, clip(r_t, 1-\\varepsilon, 1+\\varepsilon) A_t^{mask}\\right)\\right] - \\beta D_{KL}(\\pi_{\\theta,\\phi} \\,|\\, \\pi_{ref})$$
>
> Both clipping bounds matter because $A_t^{mask}$ can be positive or negative. We will add this clearer formulation and fix the Eq. 12 presentation in the revision.
>
> ---
>
> ### W5. Robustness drop under DIPPER
>
> We agree the paper should explain this drop more clearly. DIPPER-like rewriting directly perturbs the lexical and structural choices where a token-level watermark is embedded, whereas variable renaming preserves much more of the surrounding implementation pattern. In code, many seemingly small rewrites are not actually semantics-preserving once executability is enforced, so the space of strong but valid transformations is narrow.
>
> Code is also inherently more **delicate** than natural language: even small token changes (altering an operator, reordering statements) can break functionality. The set of truly semantics-preserving transformations that also preserve executability is narrow, making this a fundamentally hard attack for *any* token-level watermarking method. We will frame this as an important open challenge in the revision.
>
> ---
>
> ### W6. Missing efficiency analysis
>
> Table 2 reports parameter overhead (~7.3%) and the additional serial cost on the critical path (tensor slicing/logit adjustment only; the watermark model forward pass is overlapped with the base model). We will revise the wording to make this distinction explicit.
>
> ---
>
> ### W7. Limited evaluation scale
>
> We train on 1.5B and transfer to 8B without retraining, demonstrating policy portability within the same backbone/tokenizer family. CodeTracer's design is model-agnostic (lightweight plug-in, frozen base LLM), so the gating mechanism is independent of backbone size. We will soften the wording and state the compute limitation more directly.
>
> ---
>
> ### Q1. Why are post-hoc methods near random in Table 1?
>
> Distinguishing LLM-generated code from human-written code is inherently harder than for natural language text. Code is highly structured with strict syntactic constraints, and functionally equivalent programs can differ substantially in tokens (balance-counter vs. stack-based, `[i for i in l if i > 0]` vs. `[x for x in l if x > 0]`). Combined with short outputs (~69 tokens on HumanEval), general text detectors have very little signal to work with. We will clarify this in the revision.
>
> ---
>
> ### Q2. Metric normalization at 50%
>
> For AUROC, 50% is chance level. For TPR at fixed FPR, chance ≈ FPR (e.g., ~5% at FPR=5%), not 50%.
>
> ---
>
> ### Q3-Q4. Transferability / Runtime costs
>
> We will provide additional transfer results in the revision. For runtime, see W6 above.

---

> > ### Author Rebuttal · Reviewer_PJsj · 2026-04-02
> >
> > Note: regarding the computational cost, we already checked Table 2, but it does not compare with other methods like SWEET, WLLM, etc. Questions 3 and 4, for the time being, remain unanswered. Everything else was addressed.

---

> > > ### Author Response · Authors · 2026-04-04
> > >
> > > Thank you for the follow-up. You are right that our previous rebuttal did not make these two points explicit enough. We clarify them directly below.
> > >
> > > ---
> > >
> > > ## (Q3) Runtime latency / memory cost compared with other watermarking methods.
> > >
> > > The clearest comparison is to separate extra generation-time latency, extra generation-time memory, and detection cost:
> > >
> > > | Method | Extra generation-time latency | Extra generation-time memory | Detection cost |
> > > |---|---|---|---|
> > > | WLLM | <100 μs | negligible | low; requires the watermark key |
> > > | SWEET | <100 μs | negligible beyond the backbone already used for generation | high; requires prompt history and the backbone model |
> > > | CodeTracer | <100 μs (serial post-processing) | +118M parameters, about 0.5 GB in bf16 | low; prompt-free and backbone-free, only the watermark policy is needed |
> > > | EXP-edit | much higher | substantially higher | low once the final output is produced |
> > >
> > > Here, the <100 μs figure refers only to the additional serial post-processing time after the backbone logits are already available, i.e., the token-level tensor operation before next-token sampling. For WLLM this is PRF partitioning plus logit bias; for SWEET it is entropy thresholding from the already-available backbone logits plus logit bias; and for CodeTracer it is tensor slicing / gating / logit adjustment. The 118M watermark-policy forward pass in CodeTracer is overlapped with the backbone model, so it increases memory and total compute, but not the additional serial latency on the decoding path.
> > >
> > > Therefore, WLLM, SWEET, and CodeTracer are in the same generation-time latency regime, while differing mainly in memory footprint and detection requirements. Relative to SWEET, CodeTracer's main practical advantage is on the detection side: it is prompt-free and backbone-free, so it does not require the full backbone LLM or the original prompt history.
> > >
> > > ---
> > >
> > > ## (Q4) Why are the post-hoc methods in Table 1 close to random, unlike in their original papers?
> > >
> > > Thank you for the follow-up. We agree this point deserves a more precise explanation.
> > >
> > > **First, we included GPTZero and DetectGPT as general post-hoc references because this is also the comparison protocol used in prior code-watermark benchmarks such as SWEET [1]**. Importantly, in that same short, function-level code setting, SWEET also reports that post-hoc detectors remain weak on code benchmarks, with AUROC values close to chance on HumanEval/MBPP rather than the much stronger numbers often seen on natural-language passages. So our Table 1 should not be interpreted as an implementation failure or an unfair setup; it is consistent with prior observations in code-generation benchmarks.
> > >
> > > The main reason is that the signals used by these methods are **intrinsically weak on short, highly structured code**. GPTZero relies on perplexity / burstiness-style variation cues, while DetectGPT relies on likelihood curvature under local perturbations. On HumanEval/MBPP, completions are short function-level programs with strict syntax and many low-entropy positions. In this regime, there is little room for the distributional artifacts that generic post-hoc text detectors typically exploit.
> > >
> > > Moreover, many benchmark tasks admit a small number of canonical, concise implementations shared by both humans and LLMs. For example, in `has_close_elements`, both a nested-loop solution and a sorting-based solution are natural, executable, and task-appropriate implementations. **From a single snippet, even a human reader would have difficulty inferring provenance purely from surface form; accordingly, a generic text detector has even less reliable signal**. (For example, you can check W3 of [reply](https://openreview.net/forum?id=4xjq3iR4aK&noteId=FslJtn8Ht4) )
> > >
> > > **A second issue is scoring-model mismatch. Post-hoc detection depends strongly on the surrogate model used for scoring / perturbation, which need not match the source code model that generated the sample.** In our DetectGPT implementation, we follow the standard configuration used in prior work and in SWEET (T5-3B, 20% masking, 100 perturbations), so the weak AUROC is more plausibly explained by the code setting and surrogate-model mismatch than by any nonstandard implementation choice.
> > >
> > > [1] Who Wrote this Code? Watermarking for Code Generation, ACL 2024.

---

### Official Review · Reviewer_wpRp · 2026-03-11

**Soundness:** 3
**Presentation:** 3
**Significance:** 3
**Originality:** 3
**Overall Recommendation:** 4
**Confidence:** 3

**Summary:**

The authors introduce CodeTracer a code watermarking method where a policy model learns at each generation step whether to apply watermarking and which tokens form the green list. This model is trained with GRPO with a reward combining code execution correctness, watermark detectability and per token feedback encouraging green token selection. The evaluations are conducted on HumanEval and MBPP showing improvements over existing methods SWEET, WLLM or EXP-edit.

**Compliance With Llm Reviewing Policy:**

Affirmed.

**Final Justification:**

I am switching to Weak Accept. The rebuttal addressed my questions. However, I am not a RL specialist so I am not sure of the correctness of the approach and of the improvement compared to other methods.

**Key Questions For Authors:**

1) According to the appendix, the watermarking model is a 118M parameter transformer. I don't see how a forward pass through such a transformer can take less than 100 μs. Could you provide clarifications about this?
2) There is no ablation on the effect of the context window size for the policy model (see Weakness 1). What is the effect of this hyperparameter?
3) Which are the code patterns learned by the watermarking model? It would be nice to have an interpretability analysis.
4) Are the negative examples used for FPR measurements generated by the base LLM without watermarking or also human written code?

**Limitations:**

yes

**Strengths And Weaknesses:**

Strengths:
1) The paper is well written and using a GRPO based watermarking model is a novel way to tackle code watermarking.
2) The model is lightweight, with no need to access the full LLM at detection time and is improving over several baselines by having the highest pass @ 1 and detection performance for both HumanEval and MBPP.

Weaknesses:
1) With a context window of only two tokens, it is questionable that the watermarking model understands the structure of the code. Also, Table 6 and Appendix C3 do not seem to agree on this point with a context of 10 in Appendix C3.
2) The comparison with SWEET is not fair. You could report their performance using their setup (full generation history and original
LLM) besides the results already here with the adaptations made to their method.
3) Missing interpretability analysis on the code patterns learned by the watermarking model.
4) In Section 4, the authors propose to use KL divergence and then Table 6 sets the KL weight to 0.

---

> ### Author Rebuttal · Authors · 2026-03-31
>
> We thank the reviewer for recognizing the novelty of using a GRPO-based watermarking policy for code.
>
> ---
>
> ### W1. Context window inconsistency (c=2 vs c=10)
>
> We apologize for this inconsistency. **The actual context window used in all experiments is c=2.** The c=10 in Appendix C.3 is a drafting error from an earlier implementation description. The base LLM receives the full prefix, while the watermark policy only sees a small local window for gating and green-list construction. Our ablation on HumanEval shows the trade-off clearly:
>
> | c | Pass@1 | Pass@10 | AUROC | TPR@5%FPR |
> |:---:|:---:|:---:|:---:|:---:|
> | 1 | 66.40 | 78.09 | 62.28 | 14.02 |
> | **2** | **62.65** | **77.11** | **77.71** | **32.32** |
> | 4 | 62.65 | 75.80 | 70.96 | 21.95 |
> | 8 | 65.58 | 79.61 | 69.44 | 11.59 |
>
> c=1 and c=8 can yield slightly higher Pass@1/Pass@10, but c=2 gives substantially stronger detection (77.71 AUROC, 32.32 TPR@5%FPR) while maintaining competitive functionality. We will therefore describe c=2 as the best **utility-detectability trade-off** rather than presenting it as best on every metric. The base LLM already handles long-range dependencies; the watermark policy only needs enough local context to make gating decisions.
>
> ---
>
> ### W2. Fairness of the SWEET comparison
>
> Our main table uses the **practical detector setting** studied in our paper: prompt-free and base-model-free detection. The original SWEET detector requires the full user prompt to compute first-token entropy and access to the generating model's weights for per-token probabilities, which are unavailable in practical third-party verification.
>
> We agree it is also useful to report SWEET in its original full-information setting as a reference point. We therefore additionally ran SWEET under that setting and adapted CodeTracer (CodeTracer*) to the same information regime as a **diagnostic reference comparison**, not as a replacement for the main practical-detector method:
>
> | Method | Pass@1 | Pass@10 | AUROC | TPR@5%FPR |
> |:---|:---:|:---:|:---:|:---:|
> | SWEET (full-info) | 61.77 | 75.45 | 83.10 | 42.68 |
> | CodeTracer* (full-info) | **64.13** | **76.39** | **93.38** | **61.87** |
>
> CodeTracer* outperforms SWEET by +10.28 AUROC and +19.19 TPR even under the same privileged access. We will include both the practical-detector comparison and this explicitly labeled full-information reference comparison in the revision.
>
> ---
>
> ### W3. Missing interpretability analysis
>
> We agree that the current manuscript needs more concrete interpretability evidence. In sampled HumanEval completions, the watermark policy exploits several categories of **executable** code choices rather than comments or annotations:
>
> **Algorithm choice (`has_close_elements`):** In our sampled outputs, non-watermarked code often uses nested loops, while CodeTracer frequently selects a sorting-based approach:
> ```python
> # Non-watermarked                        # Watermarked
> for i in range(len(numbers)):            numbers.sort()
>     for j in range(i+1, len(numbers)):   for i in range(len(numbers) - 1):
>         if abs(numbers[i]-numbers[j])        if numbers[i+1]-numbers[i]
>             < threshold:                         <= threshold:
>             return True                          return True
> return False                             return False
> ```
>
> **Code organization (`get_odd_collatz`):** Non-watermarked code often uses helper-function decomposition, while CodeTracer prefers inline procedural patterns with renamed variables (`collatz_sequence` vs. `sequence`). Both pass 20/20 tests.
>
> **Algorithm alternative (`match_parens`):** Non-watermarked code uses a balance counter; CodeTracer selects stack-based validation (`stack.append`/`stack.pop` vs. `balance += 1`/`balance -= 1`).
>
> **Variable naming (`get_positive`):** CodeTracer consistently selects `x` over `i`/`num` when functionally equivalent, e.g., `return [x for x in l if x > 0]`.
>
> We will include these examples in the revision.
>
> ---
>
> ### W4. KL divergence (Section 4) but β_KL=0 (Table 6)
>
> We agree this should be stated more clearly. Eq. (12) presents the **general GRPO objective** with an optional KL regularizer; in the Table 6 sweep, the best-performing setting was **β_KL=0**, so the regularizer was effectively disabled. We will state this explicitly to avoid confusion.
>
> ---
>
> ### Q1. 118M forward pass latency (<100 μs)
>
> Table 2 reports the **additional serial overhead on the critical path** (tensor slicing/logit adjustment), not the full 118M forward pass. The watermark model runs in parallel with the base model. We agree the wording is misleading and will revise it.
>
> ---
>
> ### Q2–Q3. Context window ablation / Code patterns
>
> See W1 for the context-window trade-off and W3 for concrete code-pattern examples. We will cross-reference both points more explicitly in the revision.
>
> ---
>
> ### Q4. Negative examples for FPR calibration
>
> The negative examples are **human-written code**, not base-LLM outputs without watermarking.

---

> > ### Author Rebuttal · Reviewer_wpRp · 2026-04-03
> >
> > I thank the authors for their rebuttal. It is not clear to me why the TPR is getting worse when the context window is larger than 2.

---

> > > ### Author Response · Authors · 2026-04-04
> > >
> > > Thank you for the follow-up question. Our interpretation is that the drop in TPR for context windows larger than 2 is mainly a **watermark-signal / detectability effect**, rather than a code-understanding effect.
> > >
> > > The key point is that increasing $c$ only changes the input to the **watermark policy**. The **base code LLM still sees the full prefix**, so a larger $c$ does not provide the generator with additional global program understanding beyond what the backbone already has. This is also consistent with the ablation: Pass@k changes only modestly, while AUROC / TPR drop much more substantially.
> > >
> > > A more precise way to view this is through our detector in Eq. (2). Let $T$ be the number of scored positions, i.e., positions where the reconstructed gate predicts watermarking is active, and let $p_G(c)=\mathbb{E}[N_G/T]$ denote the average fraction of green hits among these scored positions. Then the expected z-score separation can be understood heuristically as scaling like $
> > > \frac{(p_G(c)-\gamma)\sqrt{T}}{\sqrt{\gamma(1-\gamma)}}. $
> > >
> > > Therefore, a larger context helps only if it increases the **average green-token surplus** $p_G(c)-\gamma$ enough to offset the $\sqrt{T}$ term. In our setting, this does not happen. Instead, when $c>2$, the policy becomes more context-specific and tends to activate on more ambiguous prefix states. This weakens the consistency of the watermark bias at each scored position, so $p_G(c)$ drops, and the final z-score becomes smaller even if functionality stays similar.
> > >
> > > We believe there are two concrete reasons for this:
> > >
> > > **(1) Signal dilution / context fragmentation.**
> > > With $c=2$, the policy learns a conservative, high-precision gating rule: it mainly intervenes at a small set of local positions where it can reliably push the next token toward the green list without harming code correctness. With larger $c$, the same local code pattern is split into many more distinct context states, so the learned preference becomes less consistent across occurrences. In other words, the policy fires on more specific but less repeatable contexts, which lowers the average per-token watermark effect.
> > >
> > > **(2) Generation–detection context mismatch in the prompt-free setting.**
> > > During generation, the watermark policy conditions on the last $c$ tokens of the **prompt + generated prefix** (Algorithm 1), while during detection it reconstructs decisions from the **generated code only** (Algorithm 2). Thus, as $c$ grows, more early watermark decisions depend on prompt-side context that is unavailable at detection time. This mismatch is minimal at $c=2$, but becomes more pronounced for $c=4$ and $c=8$, making reconstruction of $w$ and $G$ noisier and further hurting TPR. This matters especially on HumanEval, where many completions are short.
> > >
> > > This explanation is also consistent with the benchmark length distribution. In our HumanEval run under the main $c=2$ setting, the median number of scored tokens is **52.5**, **46.95%** of completions have at most **50** scored tokens, and **73.78%** have at most **80** scored tokens across **164** tasks. On such short sequences, even a modest reduction in $p_G(c)-\gamma$ can noticeably reduce the final z-score and therefore TPR.
> > >
> > > Empirically, this pattern is consistent with the ablation table. Relative to $c=2$, increasing the policy context to $c=4$ or $c=8$ leaves functionality similar or slightly higher (Pass@1: **62.65** at $c=2$ vs. **62.65 / 65.58**; Pass@10: **77.11** vs. **75.80 / 79.61**), but reduces detectability substantially (AUROC: **77.71** vs. **70.96 / 69.44**; TPR@5%FPR: **32.32** vs. **21.95 / 11.59**). This is exactly the pattern expected when larger contexts fail to increase the usable green surplus and instead make the watermark signal harder to reconstruct under prompt-free detection.
> > >
> > > So our claim is not that larger context is never useful for code modeling in general. Rather, for this benchmark setting and this detector design, $c=2$ provides the best **utility-detectability trade-off**:
> > >
> > > 1. $c=1$ is too short to identify enough safe watermark positions.
> > > 2. $c \ge 4$ introduces more context dependence than is helpful, which weakens reconstruction consistency and dilutes the watermark signal.
> > >
> > > We will clarify this mechanism in the revision.

---

### Decision · Program_Chairs · 2026-04-30

**Decision:**

Accept (regular)

**Comment:**

This paper receives mixed reviews. On the positive side, reviewers acknowledge the proposed method for its clarity, well-motivated design, and several careful design choices. Prior to the rebuttal, however, reviewers also raised several concerns, including the analysis and clarification of certain experimental results, transferability to other foundation models, and runtime costs. The authors provided a careful rebuttal, and after the rebuttal, reviewers wpRp, fVtj, and 2NkW remained positive about this work. While reviewer PJsj raised follow-up concerns, these mainly concerned missing additional runtime comparison results, which the authors stated in the rebuttal were being run and later included in the follow-up reply. Beyond this, reviewer PJsj did not indicate additional concerns. Therefore, the AC believes the concerns in the reviews have been resolved and recommends accept.

Meanwhile, the AC would like to alert the authors regarding a potential incorrect or hallucinated citation:
Tian, E., Wang, Y., and Dai, Z. Gptzero: An open-source initiative for ai-generated text detection. arXiv preprint arXiv:2303.08217, 2023. The title and authors mismatch with arXiv.